# Flexible Electronics: Advancements and Applications of Flexible Piezoelectric Composites in Modern Sensing Technologies

**DOI:** 10.3390/mi15080982

**Published:** 2024-07-30

**Authors:** Jinying Zhang, Jiacheng Wang, Chao Zhong, Yexiaotong Zhang, Yajuan Qiu, Lei Qin

**Affiliations:** 1Beijing Key Laboratory for Precision Optoelectronic Measurement Instrument and Technology, School of Optics and Photonics, Beijing Institute of Technology, Beijing 100081, China; 3120225346@bit.edu.cn (J.W.); 3120220661@bit.edu.cn (Y.Z.); 2Yangtze Delta Region Academy of Beijing Institute of Technology, Jiaxing 314001, China; 3Beijing Key Laboratory for Sensors, Beijing Information Science & Technology University, Beijing 100101, China; 20192289@bistu.edu.cn (C.Z.); q18804096327@163.com (Y.Q.)

**Keywords:** sensors, flexible electronics, piezoelectric composite material, precision machining

## Abstract

The piezoelectric effect refers to a physical phenomenon where piezoelectric materials generate an electric field when subjected to mechanical stress or undergo mechanical deformation when subjected to an external electric field. This principle underlies the operation of piezoelectric sensors. Piezoelectric sensors have garnered significant attention due to their excellent self-powering capability, rapid response speed, and high sensitivity. With the rapid development of sensor techniques achieving high precision, increased mechanical flexibility, and miniaturization, a range of flexible electronic products have emerged. As the core constituents of piezoelectric sensors, flexible piezoelectric composite materials are commonly used due to their unique advantages, including high conformability, sensitivity, and compatibility. They have found applications in diverse domains such as underwater detection, electronic skin sensing, wearable sensors, targeted therapy, and ultrasound diagnostics for deep tissue. The advent of flexible piezoelectric composite materials has revolutionized the design concepts and application scenarios of traditional piezoelectric materials, playing a crucial role in the development of next-generation flexible electronic products. This paper reviews the research progress on flexible piezoelectric composite materials, covering their types and typical fabrication techniques, as well as their applications across various fields. Finally, a summary and outlook on the existing issues and future development of these composite materials are provided.

## 1. Introduction

Sensors are core components of modern instrumentation that play a crucial role in detecting and transmitting signals, enabling the accurate acquisition of external information. With the continuous advancement and innovation of instrumentation systems, there is an increasing demand for the development of novel sensors [1,2,3]. Nowadays, the main types of sensors are piezoelectric, piezoresistive, capacitive, and inductive sensors. Depending on the requirements of different fields, they are utilized in various testing instruments and devices. The different types of sensors, their main application areas, and their advantages and disadvantages are shown in Table 1. Among various types of sensors, piezoelectric sensors, made from piezoelectric materials, are widely used in multiple technical fields, such as weapon guidance, industrial inspection, marine exploration, and medical diagnosis [4,5,6,7,8,9,10], due to their advantages of high sensitivity, strong temperature stability, and fast response speed [4]. A piezoelectric sensor operates on the principle of the piezoelectric effect. When a piezoelectric material is subjected to mechanical stress, it generates electric charges on its surface (positive piezoelectric effect). Conversely, when an electric field is applied to the surface of the piezoelectric material, it generates mechanical deformation (inverse piezoelectric effect). Depending on whether the direct or inverse piezoelectric effect is utilized, piezoelectric materials can be designed as either passive or active sensors. The schematic diagram of the principle of the piezoelectric effect is shown in Figure 1. Besides, piezoelectric sensors also exhibit excellent electrical response characteristics. Because piezoelectric materials can sense varying degrees of stress and strain, researchers can evaluate mechanical information through changes in the amplitude or phase of the output voltage or current. This is an important reason why these sensors have been actively used in various research fields and have developed rapidly in recent years. Figure 2 displays the response curves of two types of piezoelectric sensors to mechanical changes: (a) the relationship between various degrees of finger bending and the output voltage of the sensor [11], and (b) the relationship between the output current of sensors with different MoS_2_ doping contents under stress conditions [12]. When an external force is applied, the electrical response output by the piezoelectric sensor is quite significant. Traditional piezoelectric materials are primarily classified into three types [13,14]: piezoelectric single crystals, piezoelectric polymers, and piezoelectric ceramics. However, over time, the limitations of piezoelectric materials have become increasingly apparent. Piezoelectric single crystals, such as natural quartz, Rochelle salt, and lithium niobate, were crucial in the early development of piezoelectric sensors. However, they are characterized by high brittleness and difficult processing, making it challenging to meet the requirements for low-cost production. Piezoelectric polymers, such as PVDF and PVC, possess high flexibility and strong compatibility. Although they meet the adaptability and biocompatibility of sensors, they suffer from undesired piezoelectricity. Piezoelectric ceramics, such as lead zirconate titanate (PZT), lead titanate (PTO), and barium titanate (BTO) exhibit high electromechanical conversion efficiency and good temperature stability. However, they also face significant brittleness issues, making them unsuitable for high adaptability and low acoustic impedance environments. Therefore, to address the inherent drawbacks of traditional piezoelectric materials, researchers have developed piezoelectric composite materials, broadening their application spectrum [13,15,16,17].

Piezoelectric composite materials, formed by blending piezoelectric materials and polymers in specific proportions, originated in the late 1970s. They can be categorized into ten interconnected structures based on spatial arrangement, as depicted in Figure 3 [22]. The advantages of polymer materials, including low acoustic impedance, low density, and high flexibility, can significantly improve the shortcomings of traditional piezoelectric materials, endowing piezoelectric composite materials with numerous novel attributes distinct from those of traditional materials [14,23,24,25]. Early studies on piezoelectric composite materials primarily focused on the PZT-epoxy type, which has been extensively utilized in various domains, including underwater acoustics [26,27,28,29,30,31,32,33,34], medical diagnosis [35,36,37,38,39,40], and health monitoring [41,42,43,44]. This type of composite material is characterized by high hardness and strong electromechanical and piezoelectric properties, making it a crucial sensitive material for developing acoustic transducers [45,46]. MSI, a company in the United States, reported a curved transducer array based on the PZT-epoxy type, fabricated using thermoforming [28]. The transducer array, composed of eight structurally similar composite material elements, can meet the requirements of wide-beam transmission (>60°) and high-sensitivity reception (~178 dB) for underwater detection. Badcock et al. [47] embedded PZT powder into an epoxy resin matrix to produce a 0-3 type composite material. Upon testing, this composite material not only effectively absorbed Lamb waves but also generated a sensitivity response twice as high as PVDF. Zhang et al. [40] proposed a lateral mode 2-2 type piezoelectric composite material composed of PZT-5H and rigid epoxy resin. The material achieves an effective fluid static pressure coefficient (*d*_h_) of 6000 pC/N, making it suitable for developing underwater transducers capable of withstanding high hydrostatic pressure. Fleury et al. [38] utilized a lay-up and casting method to fabricate 1-3 type piezoelectric composite materials, extending their application to high-power diagnosis and therapy. With a high electromechanical coupling coefficient (around 0.6) and lower acoustic impedance, the transducer can generate ultrasound power up to 30 Watts/cm^2^, maintaining long-term stability at mechanical efficiencies of around 50% to 60%. This technology has significant application value for real-time ultrasound monitoring. Furthermore, researchers have reported various modifications to the structure of PZT-epoxy-base composite materials [48,49,50,51,52,53], resulting in notable enhancements in energy conversion efficiency, temperature stability, and mechanical properties. The PZT-epoxy type piezoelectric composite material, with its excellent performance, has enabled breakthroughs in acoustic transducer performance in terms of high electroacoustic response, mechanical stability, and high-power output. However, despite these many advantages, the high hardness of piezoelectric ceramics and the high rigidness of epoxy resin polymers often confine the design of PZT-epoxy type composite materials to rigid, non-bendable forms, making it difficult to exhibit flexible characteristics. This severely limits the application of these materials in the field of flexible sensors.

In recent years, a series of flexible devices have emerged to meet the demands for high precision, mechanical flexibility, and miniaturization in sensing [54,55,56,57,58,59,60]. This trend has steered the development of piezoelectric composite materials towards enhanced adaptability, sensitivity, reliability, and performance compatibility, moving away from traditional PZT-epoxy-type structural designs. Huang et al. [61] proposed a porous PVDF polymer with high piezoelectricity. This material is composed of traditional polyvinylidene fluoride and barium titanate, compressed and molded to demonstrate high flexibility and excellent piezoelectric properties. Experimental findings indicate that the material can increase *d*_33_ (piezoelectric strain coefficient) to 51.20 pC/N and achieve a high sensitivity of 132.87 mV/kPa, offering a novel pathway for developing self-powered flexible sensors. Jiang et al. [62] reported a flexible composite material synthesized from PZT and silicone resin, with Young’s modulus of 34.2 MPa, exhibiting greater flexibility than traditional PVDF. Upon testing, this composite material showed a yield failure strain of 7.2% and a fatigue failure strain of 9200 με, demonstrating significant potential for impact damage monitoring applications. Chen et al. [63] prepared a carbon nanotubes (CNT)/polyvinyl alcohol (PVA)/zinc oxide (ZnO) composite material. Under pulse loading, this composite material displayed piezoelectric sensitivity, linear stability, and response accuracy of 4.87 mV/lbf, 3.42%, and 1.496 ms, respectively, demonstrating rapid response and high-precision sensing features. The development of flexible sensors has been driven by advances in flexible piezoelectric composite materials, rapidly expanding into various fields such as underwater detection, wearable sensors, flexible electronic skins, targeted therapy, and ultrasound diagnostics for deep tissue. Therefore, the research and development of flexible piezoelectric composite materials is of paramount importance. This paper reviews the research progress of flexible piezoelectric composite materials, enumerates their types and common fabrication methods, outlines their applications in various domains, and finally, discusses future development directions.

## 2. Types of Flexible Piezoelectric Composite Materials

Flexible piezoelectric composite materials are a type of composite that balances mechanical stretchability and piezoelectricity, serving as the core component of flexible sensors. Based on their connectivity structure, flexible piezoelectric composite materials are primarily categorized into 0-3 and 1-3 types. In recent years, flexible sensors have gained increasing attention in the medical field. This is mainly due to the advantages of flexible piezoelectric composite materials, such as high biocompatibility, high flexibility, and high sensitivity. However, since these sensors often need to be implanted in human tissue or attached to human skin, it is also important to develop flexible piezoelectric composite biomaterials that are biodegradable and non-toxic.

### 2.1. 0-3 Type

0-3 types of flexible piezoelectric composite materials [15,64] are two-phase materials that are composed of zero-dimensional piezoelectric particles and three-dimensional flexible polymers, typically in the form of thin films. These composite materials integrate the piezoelectricity of these particles with the flexibility of polymers, making them highly popular in current research on flexible sensing. Regarding material composition, the two constituent phases serve distinct roles: piezoelectric particles are usually made from highly piezoelectric powders to ensure strong force-electric response characteristics while the polymer matrix needs to be highly flexible and stretchable to provide the material with a high strain capacity. 0-3 type flexible piezoelectric composite materials are favored for their high sensitivity, good flexibility, and strong compatibility, holding significant practical value in developing the next generation of flexible electronic devices [65,66]. This paper categorizes 0-3 type flexible piezoelectric composite materials into PVDF series matrix type and non-PVDF series matrix type based on whether the polymer matrix possesses piezoelectricity.

#### 2.1.1. PVDF Series Matrix Type

Polyvinylidene fluoride (PVDF) is a common polar crystalline piezoelectric polymer. It has become an important material for developing flexible sensors due to its high flexibility and compatibility [67,68,69]. However, the piezoelectricity of PVDF is relatively weak. Researchers typically enhance its piezoelectric properties by doping the PVDF matrix with highly piezoelectric particles, creating PVDF-based 0-3 piezoelectric composite materials. This method addresses the low piezoelectricity of pure PVDF and is the mainstream modification approach, which has been extensively reported in the field of flexible piezoelectric composites. Additionally, PVDF derivatives such as polyvinylidene fluoride-trifluoroethylene (P(VDF-TrFE)) and polyvinylidene fluoride-chlorotrifluoroethylene (P(VDF-CTFE)) are similarly modified.

Ail et al. [70] electrospun ZnO@ZnS nanoparticles with PVDF slurry to create a PVDF-based nanocomposite material, combining flexibility and high piezoelectricity. The incorporation of ZnO@ZnS nanoparticles facilitated the transformation to the electroactive β-phase in PVDF, greatly improving the piezoelectric properties of the composite material. Experimental results showed that the conversion sensitivity of conventional PVDF devices is around 0.091 V/N·mm^3^, whereas this composite material device achieves a conversion sensitivity of 0.153 V/N·mm^3^, an increase of nearly 70%. This demonstrates that the enhancement in PVDF’s piezoelectric properties is closely related to the optimization of the device’s sensing characteristics. Talab et al. [71] reported a BTO/P(VDF-TrFE) composite material. Due to the incorporation of BTO ceramic powder, this composite material simultaneously enhances mechanical properties and piezoelectricity. Experiments showed that this composite material has a *g*_33_ value (piezoelectric voltage constant) of 205 mV·m/N and a polarization value of 6.18 μC/cm^2^, representing increases of approximately 30% and 46%, respectively, compared to pure P(VDF-TrFE), demonstrating significantly enhanced inverse piezoelectricity. Furthermore, researchers explored the sensing properties of this composite material. Figure 4 shows the voltage response curves of BTO/P(VDF-TrFE) and P(VDF-TrFE) materials in the *g*_31_ mode. As the finger varies under low, moderate, and high bending conditions, the output voltage of the BTO/P(VDF-TrFE) sensor is almost twice that of the P(VDF-TrFE) sensor. Meanwhile, the output power is 4.6 times that of the P(VDF-TrFE) sensor, fully demonstrating the application value of this flexible composite material in sensing, actuation, and energy harvesting devices. Xiong et al. [72] combined BTO nanoparticles with P(VDF-CTFE) to create a BTO/P(VDF-CTFE) composite material with high energy storage characteristics, as shown in Figure 5. The composite material employs a layered design, with the top and bottom layers of P(VDF-CTFE) polymer effectively enhancing the composite’s breakdown field strength and dielectric constant. The middle layer consists of dopamine-modified and BTO-doped BTO/P(VDF-CTFE) film, improving charge-discharge efficiency and polarization strength. Experiments have shown that this layered BTO/P(VDF-CTFE) composite material has a dielectric constant of 10.44, a maximum breakdown field strength of 362.25 kV/mm, and a charge-discharge efficiency of 86.63% under an electric field strength of 500 kV/cm, indicating excellent dielectric and energy storage properties.

Besides enhancing piezoelectricity, some studies also aim to improve other physical properties of PVDF to further enhance device functionality and sensing characteristics. Traditional PVDF exhibits poor melt processability and cannot withstand high voltage, significantly limiting its application in devices. To address this issue, He et al. [73] proposed a PVDF-PLLA-C1 nanocomposite material. By incorporating small amounts of crystalline polymer (PLLA) and fluorescent material (Tb(TTA)_3_(TPPO)_2_), they significantly promoted the melting and crystallization behavior of the PVDF matrix. Experiments showed that this composite material sensor could produce a sensitivity response four times higher than that of traditional PVDF sensors, with an output voltage of up to 2.5V. Additionally, it can absorb ultraviolet light and emit visible light, making it suitable for developing highly responsive force-sensitive sensors and self-powered conformal devices.

In summary, PVDF series matrix-type composite materials generally optimize their piezoelectric properties through doping, often using a 0-3 type composite structure (doped phase-PVDF matrix phase). The greatest advantage of these composite materials is that they retain the inherent high conformability and high compatibility of PVDF materials, offering significant potential for developing high-strain, high-sensitivity sensors.

#### 2.1.2. Non PVDF Series Matrix Type

Another type of 0-3 flexible piezoelectric composite material is composed of highly piezoelectric particles and non-piezoelectric polymers. Since the polymer primarily provides mechanical stretchability and deformability, the piezoelectricity of this 0-3 type composite material comes entirely from the doped piezoelectric particles. Compared to the PVDF series matrix type, non-PVDF matrix type composite materials might exhibit a lower piezoelectric coefficient. To balance the issue of piezoelectricity, researchers typically select piezoelectric ceramic powders with high piezoelectric coefficients that are also easy to manufacture as the piezoelectric phase material for the composites.

Jiang et al. [74] combined PZT powder with silicone resin to prepare a highly flexible 0-3 PZT/silicone resin composite material, as shown in Figure 6a. They conducted studies on the effects of polarization conditions and PZT structural parameters on the piezoelectric properties of the composite material. Experiments demonstrated that the piezoelectric strength of the 0-3 PZT/silicone resin composite material is closely related to the polarization conditions. The optimal sensing sensitivity was achieved when the polarization time was 25 min, the polarization field was 4 kV/mm, the polarization temperature was 100 °C, the PZT doping volume fraction was 50%, and the powder particle size was 170–212 μm. Moreover, within the environmental temperature range of −55 to 75 °C, the 0-3 type sensor exhibited excellent flexibility and sensitivity in detecting guided wave signals. Bradley et al. [75] mixed PZT powder with fluorinated suspensions and used screen printing, template printing, and dip-coating techniques to produce a 0-3 PiezoPaint composite material with a wideband acoustic response. Compared to 1-3 type and 2-2 type composite materials, this 0-3 type material has a thinner thickness and smaller size (thickness 50 μm, area 1 mm^2^), making it easy to integrate into medical devices such as catheters and probes. The team proposed a structure of an integrated photoacoustic probe comprising 0-3 PiezoPaint composite material and a fiber Bragg grating (FBG). This probe features high frequency, small size, and high flexibility, capable of achieving pulse-echo responses of up to 20 MHz and radiofrequency responses at common MRI frequencies. JO et al. [76] developed a PZT/PVB 0-3 type composite material using PZT powder and polyvinyl butyral (PVB). The high thermoplasticity of the PVB material enables the 0-3 composite material to maintain excellent piezoelectric performance even at high temperatures. Additionally, they fabricated this composite material into a force-sensitive sensor, as shown in Figure 6b. Experimental results demonstrated that this sensor had sensitivities of 1.368 V/N and 0.815 V/N on aluminum oxide and polyester substrates, respectively, with response times within 45 ms, indicating high sensitivity and fast response characteristics.

The doping of a certain amount of piezoelectric ceramics significantly enhances the activity of 0-3 composite materials, unleashing the powerful potential of these materials to perceive weak signals. Recent research has uncovered the following features regarding the application of 0-3 piezoelectric composite materials: firstly, these materials are typically used in the form of small-sized films with a thickness of less than 1 mm, making them mostly suitable for developing high-frequency micro-sensing series of piezoelectric sensors; and, secondly, 0-3 piezoelectric composite materials generally exhibit high mechanical flexibility, with some also demonstrating excellent high-temperature resistance. Nevertheless, a direct trade-off exists between the piezoelectricity and flexibility of these composite materials. With an increase in the content of piezoelectric ceramics, the piezoelectric coefficient of 0-3 composite materials will increase significantly, while flexibility will decrease accordingly. To achieve the optimal balance between flexibility and piezoelectricity, it is imperative to avoid excessively high doping amounts of piezoelectric ceramic powder, thus ensuring that the piezoelectric strain coefficient remains within the application range of tens of pC/N. Hence, the advancement of piezoelectric materials with enhanced piezoelectric properties and improved compatibility can significantly enhance the performance and expand the application scope of 0-3 type piezoelectric composite materials.

### 2.2. 1-3 Type

Among the ten connectivity structures, 1-3 piezoelectric composite materials [4,13,77,78,79] are the most prevalent and extensively utilized type, celebrated for their distinctive thickness vibration mode, low acoustic impedance, and high piezoelectric coefficient. They are widely employed in the realms of underwater acoustic detection and ultrasound imaging. This involves embedding one-dimensional linear arrays of piezoelectric materials into a three-dimensional polymer matrix composite. Conventionally, piezoelectric ceramics and epoxy resins are utilized in composite fabrication, yielding predominantly planar structures with limited flexibility. Recently, there has been a continuous rise in demand for curved sensors [80,81,82,83]. In response, researchers have proposed combining piezoelectric ceramics with flexible polymers, resulting in the development of a series of highly adaptable 1-3 flexible piezoelectric composite materials. Moreover, the electromechanical properties of traditional 1-3 type composite materials have been enhanced due to the capability of flexible polymers to promote the thickness vibration of piezoelectric materials.

Kim et al. [84] fabricated a 1-3 flexible piezoelectric composite material comprising PZT-5H ceramics and polydimethylsiloxane (PDMS). This design combines the robust piezoelectric properties of the ceramics with the high mechanical flexibility of PDMS, achieving a thickness electromechanical coupling coefficient (*k*_t_) of the 1-3 composite material of up to 0.74, comparable to the length electromechanical coupling coefficient (*k*_33_) of PZT-5H, along with an approximately 50% −6 dB bandwidth. Serving as the active phase, PDMS effectively mitigated the planar coupling observed in epoxy resin in traditional 1-3 composite materials, demonstrating robust flexibility in the thickness vibration of the piezoelectric ceramics and minimizing the loss of vibration energy. Likewise, Hou et al. [85] utilized PDMS to fabricate PZT-8/PDMS 1-3 flexible composite material, depicted in Figure 7 illustrating the three-dimensional structure of this composite. The transducer made from this composite material exhibits robust mechanical flexibility and exceptional acoustic performance in terms of frequency (~1.51 MHz) and electromechanical coupling coefficient (*k*_t_ ~ 0.74).

In addition to their applications in medical ultrasound, 1-3 flexible piezoelectric composite materials play a crucial role in underwater acoustics engineering. To address the low mechanical flexibility, Hao et al. [86,87] employed a combination of the dice-fill method and island bridge electrode structure to fabricate a PZT-5A/rubber 1-3 flexible underwater transducer, depicted in Figure 8. Unlike PDMS material, the silicone rubber material can achieve a reliable connection of piezoelectric elements through flexible electric bridges, eliminating the need for complex nano silver wire electrode preparation techniques. Tested results showed that the PZT-5A/rubber 1-3 flexible composite material can bend with curvature radii ranging from 100 mm to 200 mm, offering valuable insights for developing a new generation of underwater transducers with enhanced sensitivity and flexibility.

1-3 flexible piezoelectric composite materials, characterized by their unique advantages of high electromechanical properties, which are known for their piezoelectricity and exceptional flexibility, have paved the way for the advancement of next-generation flexible and curved surface sensing devices. The two constituent phases of 1-3 flexible piezoelectric composite materials serve distinct roles: piezoelectric ceramics are commonly employed in the piezoelectric phase, offering high piezoelectric coefficients and facilitating thickness vibration modes, while materials such as PDMS and silicone rubber are preferred in the polymer phase due to their high flexibility and robust decoupling ability, enabling the composite material to achieve bending capabilities and enhance its electromechanical conversion efficiency. According to this design approach, meticulous consideration of the selection of the two constituent phases will be crucial for the development of high-performance 1-3 flexible piezoelectric composite materials and their sensors.

### 2.3. Special Structural Design

Besides typical structures like 0-3 and 1-3 types, certain flexible composite materials prioritize characteristics such as piezoelectricity, electromechanical properties, stability, and compressive strength. This makes the developed flexible piezoelectric composites more suitable for specific applications, showing significant prospects for advancement. However, due to increasingly diverse fillers, these composite materials may exhibit relatively complex multiphase interconnected structures.

Improving the piezoelectric properties has always been an important topic for PVDF series polymers. However, the enhancement of the piezoelectric performance of these polymer materials is not limited to doping with high-piezoelectric powders. Particularly in recent reports, the rational doping of certain two-dimensional conductive fillers has also been proven to effectively enhance the piezoelectricity of PVDF series polymers. Wang et al. [88] combined a two-dimensional conductive filler known as MXenes with P(VDF-TrFE) polymer, creating an enhanced composite fiber sensor with high piezoelectricity. Figure 9a shows the structure of this enhanced composite fiber and the sensor configuration. They found that MXene not only has a large number of surface functional groups that can interact with the dipoles of P(VDF-TrFE) molecular chains but also has high conductivity, which can increase the polarization of PVDF-TrFE during electrospinning. Compared with pure P(VDF-TrFE), the MXene-doped P(VDF-TrFE) sample exhibited higher output voltage and current amplitude under the same pressure stimulus, resulting in better piezoelectric response, as shown in Figure 9b. Similarly, Sharma et al. [89] also prepared this type of MXenes and P(VDF-TrFE) polymer-enhanced composite fibers based on electrospinning technology, as shown in Figure 10. The large surface area and strong adhesion of MXenes ensure a more uniform internal structure of the piezoelectric polymer. Experimental results showed that this doped enhanced composite film has an extremely low detection limit of 1.5 Pa, a wide pressure range of up to 400 kPa, and excellent stability (achieving over 10,000 cycles under high pressure of ~167 kPa). It not only has good reliability but can also be used to develop pressure sensors for detecting large deformations. In addition to MXenes, related researchers have demonstrated that the use of graphene also significantly enhances the output voltage and power of P(VDF-TrFE) [90]. Thus, it can be seen that two-dimensional conductive fillers also have significant value in improving PVDF series polymers. In addition to high conductivity, these two-dimensional conductive fillers also possess notable high mechanical stability, dielectric properties, and electrochemical characteristics, providing important insights for the development of high-performance piezoelectric composite fibers.

To achieve a balance between compression resistance and flexibility, Qin [91] introduced a large-scale 1-3 piezoelectric composite material array with a binary polymer structure, illustrated in Figure 11a. This piezoelectric composite material array consists of several identical structures of 1-3 PZT/epoxy units spliced together. The strong rigidity of epoxy resin enhances the stability and compression resistance of each unit. Silicone rubber material connects each 1-3 PZT/epoxy unit. With its high flexibility, silicone rubber significantly suppresses the lateral coupling effect and adds flexibility to this composite structure, thereby enhancing the piezoelectric coefficient and adaptability of the material. To maintain both compression resistance and flexibility, Wang et al. [92] introduced a nested piezoelectric composite material with semi-flexible characteristics, depicted in Figure 11b. This composite material utilizes both rigid and flexible materials as filling polymers, providing the composite material with axial compression resistance and unidirectional bending capability, making it suitable for developing horizontally wide-beam directional curved transducers. The designs of the aforementioned flexible composite materials, while retaining the stability advantages of traditional 1-3 types, have significantly enhanced the adaptability of the materials by introducing a third phase of silicone rubber to increase material decoupling and shape adaptability.

To address dielectric and mechanical mismatches, Tang et al. [93] utilized a (P(VDF-TrFE-CFE) copolymer doped with PZT, PDMS, and CNTs to develop a 3-3-3 piezoelectric composite material. Initially, PZT and PDMS were connected in a 3-3 configuration, forming a stable interconnected PZT framework within the PDMS matrix. Subsequently, this interconnected PZT framework was embedded into the (P(VDF-TrFE-CFE) doped with CNTs, resulting in a 3-3-3 structure. Here, PZT acts as the interconnected structure, effectively transmitting load paths during the electromechanical coupling process, whereas (P(VDF-TrFE-CFE) serves as a transition layer for adjusting the electric field distribution within the PZT framework. Figure 12 illustrates the exceptional piezoelectric performance of the 3-3-3 composite material, with a measured piezoelectric coefficient of up to 250 pm/V, an electromechanical coupling coefficient of 0.65, an ultra-low acoustic impedance of 3MRyl, and high cyclic stability under 50% compression strain. The synergistic integration of ultra-high piezoelectric performance and superior flexibility holds promise for advancing the emerging applications of flexible smart electronic products.

### 2.4. Flexible Piezoelectric Composite Biomaterials

Compared to lead-containing piezoelectric ceramics, which are high in lead content and prone to pollution, flexible piezoelectric composite biomaterials that balance piezoelectricity and safety are highly favored by medical researchers. The development of these composite materials holds profound significance for a range of biosensors suitable for biological detection or treatment. Firstly, these materials offer broad application potential in the biomedical field, such as in implantable medical devices and biosensors. Their degradability and non-toxicity allow for safe degradation within the body, thereby reducing the side effects of implants on the human body and lowering the risk of secondary surgeries. Secondly, their inherent piezoelectric properties enable the conversion of mechanical energy into electrical energy, making them suitable for self-powered devices and energy harvesting systems, thus enhancing the intelligence and sustainability of medical equipment. Furthermore, the development of these materials promotes green technology, reduces environmental pollution, and aligns with the global trend of sustainable development. Therefore, the research and development of flexible piezoelectric composite biomaterials are not only poised to bring breakthroughs in both technological and medical fields but also to foster positive advancements in environmental protection and health.

Yang et al. [94] proposed a flexible composite film based on γ-glycine crystals embedded in a PVA-glycine-PVA structure, as shown in Figure 13a. In this film, strong hydrogen bonds can form between the oxygen atoms in glycine and the hydroxyl groups on the PVA chains, leading to the nucleation and growth of γ-glycine as well as the oriented alignment of electric domains. This structure endows the film with excellent piezoelectric properties and biocompatibility. Testing revealed that the *g*_33_ of the PVA-glycine-PVA film reached 157.5 × 10^−^^3^ Vm/N, demonstrating piezoelectric performance comparable to that of commercial PVDF. This makes it suitable for developing transient implantable electronic devices. Chorsi et al. [95] combined polycaprolactone (PCL) with glycine to create a stable piezoelectric composite fiber material with an ultrasonic output reaching up to 334 kPa, as illustrated in Figure 13b. Each fiber in this composite material consists of uniformly dispersed glycine particles within a PCL matrix. The piezoelectric properties of glycine, combined with the flexibility of the PCL matrix, give this composite excellent electromechanical coupling and mechanical flexibility. Studies have shown that this composite material’s ultrasonic transducers can be effectively used for glioblastoma (GBM) mediated therapy. Among various control groups, the combined treatment mediated by the glycine-PCL transducer exhibited the best anti-GBM activity, significantly extending the survival time of mice (up to 72 days) and demonstrating its potential application in medical auxiliary treatments. Wang et al. [96] fabricated a bio-composite film made of PVA and glycine using electrospinning, as shown in Figure 13c. This film not only possesses excellent piezoelectric properties but also has degradable and self-charging capabilities. During breathing, the airflow deforms the composite material, generating charges through the piezoelectric effect, which enhances the filter membrane’s ability to adsorb fine particles. Experiments showed that masks made from the glycine-PVA composite film achieved a filtration efficiency of 97.1% for 0.3 μm particles, providing significant respiratory protection. Additionally, degradation experiments indicated that these masks could biodegrade in soil within approximately six weeks, effectively preventing environmental pollution.

Therefore, glycine, as a crystal with unique properties, holds significant advantages in the development of piezoelectric composite biomaterials. Due to the piezoelectricity, non-toxicity, and biodegradability of glycine crystals, these bio-composite materials have made many breakthroughs in the field of implantable medical devices. The appropriate selection of piezoelectric phase materials is a necessary prerequisite for developing high-performance, compatible piezoelectric composite biomaterials. According to the existing literature, piezoelectric glycine is often used as the piezoelectric phase in piezoelectric composite biomaterials. Glycine’s uniqueness lies in its polymorphic nature as an amino acid crystal, primarily existing in α, β, and γ crystal structures [97]. Among these, the α form is non-piezoelectric, while the β and γ forms are noted for their high shear piezoelectricity and thickness piezoelectricity, respectively. However, inducing and preparing these glycine crystal structures require stringent technical conditions, such as specific thermodynamic and kinetic conditions. To address this challenge, Zhang et al. [98] proposed an active self-assembly strategy to prepare β-glycine films using electrohydrodynamic spray methods, as shown in Figure 14. The synergistic effects of nanoscale confinement and in-situ polarization cause the crystal grains to align in the strongest polarization direction, resulting in films with excellent uniform piezoelectricity. Studies found that the *d*_33_ of the β-glycine film is 11.2 pm/V, and *g*_33_ is 252 × 10^−^^3^ Vm/N. Additionally, the thermal stability of these films significantly improved under the nanoscale confinement effect, reaching up to 192 °C. This approach addresses the long-standing challenge of synthesizing large-scale high-performance piezoelectric biomaterials. Beyond glycine crystals, researchers have discovered high piezoelectricity in molecular crystals such as HOCH_2_(CF_2_)_3_CH_2_OH [2,2,3,3,4,4-hexafluoropentane-1,5-diol (HFPD)], with a *d*_33_ of approximately 138 pC/N and *g*_33_ up to 2450 × 10^−^^3^ Vm/N, also demonstrating good biocompatibility and safety for biological cells [99]. Moreover, certain biological tissues with special structures can be used to develop highly compatible piezoelectric biomaterials. The layered structure of the small intestinal submucosa (SIS) has been confirmed to possess weak piezoelectricity. Zhang et al. [100] used van der Waals exfoliation to prepare an ultra-thin SIS film and employed PFM technology to reveal the bio-piezoelectricity in SIS, with a shear piezoelectric coefficient of 4.1 pm/V, showing great potential in developing biosensors, as shown in Figure 15. These piezoelectric preparation methods are of profound significance for the development of high-performance piezoelectric composite biomaterials, significantly advancing the progress of bio-device development.

## 3. Fabrication Process of Flexible Piezoelectric Composite Materials 

### 3.1. Fabrication Process

#### 3.1.1. Molding by Hot Pressing

The fundamental concept of hot pressing molding [101,102] involves melting the polymer into a solution in a high-temperature environment, then blending it with piezoelectric material powder, and subsequently mechanically shaping it. Finally, the formed material needs to undergo polarization before it can be used. This processing method offers the advantages of one-time molding, pore-free material, and suitability for industrial production. Currently, it is widely employed in the preparation of 0-3 series flexible piezoelectric composite materials, as illustrated in the preparation process depicted in Figure 16. Furthermore, due to variations in melting points among different polymers, this process offers a high degree of temperature selectivity. Wang et al. [103] fabricated 0-3 rubber-based damping composite materials using hot pressing molding and assessed various properties, including cross-sectional morphology and piezoelectric strain coefficient. Figure 17 shows the micrographs of 0-3 rubber-based damping composite materials at different magnifications. The experiments demonstrated that the PZT piezoelectric ceramic powder was well dispersed in the rubber matrix material, exhibiting good compatibility. Moreover, the piezoelectric effect of PZT significantly contributed to enhancing the damping performance of 0-3 composite materials. Utilizing this method to prepare flexible piezoelectric composite materials offers the advantages of simplicity and high yield. However, there may be instances of piezoelectric particle agglomeration, potentially impacting the piezoelectric performance of the composite material.

#### 3.1.2. Electrospinning

Electrospinning is a process that employs an electric field to produce nanofibers from polymer solutions. This technique has matured with continuous improvements and technological advancements [104,105,106]. Electrospinning is suitable for producing fiber-based piezoelectric composite material films and is also crucial for developing highly flexible nanogenerators. Figure 18 illustrates the schematic diagram of an electrospinning apparatus [107], consisting of three components: a high-voltage power supply module, a syringe pump module, and a collector module. In the electrospinning process, the high-voltage power supply is connected to both the syringe pump module and the collector module to create an electric potential difference. When the spinning solution exits the needle, a droplet forms at the tip, which deforms under the influence of electrostatic forces from the electric field. As the electric field strength increases, the droplet gradually forms a Taylor cone before being deposited onto the collector plate. During flight, the jet is also subjected to the electric field force. Accelerating flight decreases the jet diameter, and the solvent in the jet evaporates to form fibers during flight, which are ultimately collected by the receiving system. It is worth noting that according to Su [107], electrospinning is influenced by six factors: electric field strength, infusion rate, distance to the receiving end, roller speed, polymer concentration, solution viscosity, and environmental factors. These factors directly affect fiber structure size, product quality, and spinning efficiency. Chen et al. [108] developed a PVDF/PDMS composite film using electrospinning technology, as shown in Figure 19. This composite film can withstand various forms of deformation, such as stretching and bending while maintaining high flexibility and exhibiting excellent electromechanical conversion efficiency. Experimental results revealed that this composite film possesses an outstanding electrical response, with a maximum power density of up to 286 mW/m^2^. Moreover, after 5000 test cycles, the composite film showed no significant damage or interlayer separation, demonstrating its durability and stability. Luo et al. [109] reported a sandwich-structured piezoelectric nanocomposite film with high voltage and sensitivity. This film was fabricated by electrospinning a mixture of P(VDF-TrFE), tin oxide (SnO_2_) nanoparticles, and graphene (GR). The rational doping of conductive fillers facilitated the synthesis of the β phase in P(VDF-TrFE), endowing the composite film with superior piezoelectric properties. Tests showed that the sensor made from this composite film achieved a maximum instantaneous output power of 64.578 μW and an open-circuit voltage peak of 22.43 V. Additionally, this sensor can charge a capacitor, enabling it to power a commercial LCD thermometer or approximately 80 LED lights for about 1.6 s, providing a potential solution for the development of self-powered devices.

#### 3.1.3. Electrospray Deposition

Electrospray deposition is a material preparation technique that uses voltage to atomize a solution or suspension into fine droplets, which are then deposited onto a substrate. By controlling conditions such as the electric field and flow rate, this method can precisely form uniform thin films and nanoparticles. It offers high flexibility, uniformity, and adaptability. Similar to the electrospinning method mentioned, both techniques control the deposition behavior of liquids through an electric field. The electrospray deposition process is relatively simple and consists of several steps: First, dissolve the polymer matrix material in a solvent and add piezoelectric particles, ensuring they are evenly dispersed in the mixture. Next, load the prepared solution into the syringe of the electrospray apparatus, adjusting the instrument parameters to meet the requirements for thin film deposition. Then, activate the electrospray device; under the influence of an external electric field, the liquid forms a Taylor cone and is ejected as fine droplets, allowing the material to be uniformly deposited in the designated area. A schematic diagram of the electrospray deposition principle is shown in Figure 20 [110].

One major advantage of this technique is its ability to achieve uniform and large-area material preparation, making it highly promising for developing composite material films. Li et al. [110] utilized electrospray deposition to create a hierarchically interconnected piezoelectric composite textile (HIPCT) composed of PZT ceramic and P(VDF-TrFE), as shown in Figure 21a. This HIPCT exhibits high flexibility and can bend around a pen. The researchers tested the HIPCT’s superior voltage response characteristics, as illustrated in Figure 21b. Under a pressure of 2.2 MPa and a frequency of 30 Hz, the HIPCT achieved a power density of approximately 200 μW cm^−^^2^ at around 1 MΩ. Furthermore, under a compressive load excitation of 1 MPa and 30 Hz, the HIPCT could charge capacitors of 4.7 μF and 10 μF to 1.4 V and 0.7 V, respectively, within 60 s, fully demonstrating the practical value of HIPCT in developing energy harvesting sensors. 

Additionally, electrospray deposition can be used to prepare thermal interface materials. Avcıoğlu et al. [111] employed this method to fabricate an aluminum nitride (AlN)-PVA composite and tested its thermal properties. Figure 22 shows thermal images of three samples with different AlN-PVA spray durations on graphite foil after heating for 70 s. The “Uncoated Substrate” represents an aluminum alloy substrate without any coating, while “AlN-15” and “AlN-30” represent samples sprayed with AlN-PVA material for 15 and 30 min, respectively. The study found that aluminum alloy substrates coated with AlN-PVA material exhibited higher temperatures, which would promote better heat penetration and diffusion in the aluminum alloy substrate. The AlN-15 sample showed the highest thermal diffusivity, reaching up to 38.767 mm^2^/s, which is nearly a 30% improvement compared to the uncoated aluminum alloy substrate (26.638 mm^2^/s), confirming the value of AlN-PVA in thermal interface applications.

#### 3.1.4. Dice-Fill

The dice-fill method can be used to prepare 1-3 types of flexible piezoelectric composite materials, which not only have a simple process and short processing cycle but also can achieve large-sized composite material samples. The basic steps of this process are as follows: First, the polarized piezoelectric material is cut along parallel and perpendicular directions to form a cutting framework. Second, the cutting framework is washed with chemical reagents such as ethanol and dried. After that, a certain amount of flexible polymer solution is uniformly injected into the cutting framework, followed by vacuum curing. Then, the cured preliminary product is cut into a base and polished. Finally, metal electrodes are placed on the upper and lower surfaces of the sample to complete the fabrication of the composite material, as shown in the schematic diagram of the specific process in Figure 23a. Given that the cutting gap is directly generated by the blade, the dice-fill method has a high requirement for the accuracy of the blade. Figure 23b shows the PZT-rubber 1-3 type piezoelectric composite material fabricated by the dice-fill method [87]. Due to the integral segmentation and shaping of the entire piezoelectric ceramic, each piezoelectric unit has highly similar structural characteristics, thus ensuring a high level of consistency. Additionally, flexible polymers such as rubber can meet the bending capability of the composite material, making it highly suitable for the development of flexible array transducers. Hao et al. [87] pointed out that the reliability of the electrodes is a key factor affecting the 1-3 flexible piezoelectric composite materials. For 1-3 flexible piezoelectric composite materials, traditional rigid electrode layers are prone to breakage during bending, making it difficult to adapt to the material’s bending deformation. Therefore, the development of adaptive flexible electrodes to meet the interconnection of piezoelectric units is also a key factor affecting the material’s performance. 

#### 3.1.5. Injection-Molding

The injection-molding method involves the preparation of piezoelectric composite materials through ceramic injection, which is suitable for processing fine piezoelectric ceramic fiber composites [13,15,112,113]. The basic idea is to inject a mixture of a slurry of piezoelectric ceramics and adhesive powder into the corresponding mold. After drying and burning off the mold, inject a certain amount of polymer. Then, polish the finished product, and plate electrodes on the upper and lower surfaces of the material, finally obtaining the material product. The yield of the material produced by this method is relatively high, and the shape and structure of the piezoelectric phase can be controlled by the mold. However, in recent years, the literature on this process has mostly focused on 1-3 PZT/epoxy composite materials [114,115,116], indicating potential for the development of 1-3 flexible series piezoelectric composite materials. The specific process of injection molding is shown in Figure 24. Figure 25 shows the piezoelectric ceramic micro-pillar structures with high aspect ratios prepared by Park et al. [112] using injection molding. They found that the injection speed was crucial for the successful formation of the ceramic micro-pillars. Based on rheological analysis, the injection molding was carried out at a speed of 140 mm/s, ensuring that the microcavities in the mold were completely filled with the raw material.

#### 3.1.6. Freeze-Casting

The freeze-casting method, also known as freeze alignment technology, is a material preparation process that uses the anisotropic solidification behavior of solvents in an oriented temperature field as a template to control the shaping of piezoelectric particles in a slurry. This technique is commonly used to develop piezoelectric ceramics with porous structures, offering advantages such as controllable porosity, high flexibility, and cost-effectiveness. Figure 26 illustrates the general process of preparing flexible porous Barium Calcium Zirconate Titanate (BCZT)/PDMS composites using the freeze-casting method [117]. First, a mold designed for freeze-casting is created to shape the BCZT ceramics. Next, a suspension of BCZT ceramics and water is poured into a PDMS mold, which is then placed in a liquid nitrogen container to be cooled and frozen. The suspension freezes from the bottom up, and the solvent solidifies into a porous structure due to its anisotropic properties. Once the solvent sublimates, the porous BCZT ceramic structure is formed. Finally, the BCZT ceramics are polarized, and the PDMS polymer is filled into the ceramic pores. Electrodes are prepared on the top and bottom surfaces of the sample, completing the preparation of the flexible BCZT/PDMS composite material.

Figure 27a shows a sample of the BCZT/PDMS composite material prepared using the freeze-casting method. Yan et al. [117] pointed out that the piezoelectric composite materials prepared by this method exhibit a high piezoelectric coefficient due to the dense porous structure in the composite, which effectively transfers applied stress to the piezoelectric phase. Tests have shown that the BCZT/PDMS composite material can achieve a maximum *d*_33_ of 434 pC/N. Additionally, with a porosity of 60 vol%, the piezoelectric composite material demonstrates a maximum power density of 96.2 μW/cm^2^, showcasing its excellent piezoelectric response and potential for use as an energy harvesting device. Xie et al. [118] used the freeze-casting method to fabricate a layered PZT structure and combined it with a PDMS matrix to create a flexible PZT/PDMS piezoelectric composite material, as shown in Figure 27b. They found that when the porosity reached 50 vol%, the *d*_33_ of the PZT/PDMS composite material was as high as 750 pC/N, nearly 110% of that of dense ceramics. Furthermore, the material maintained a high *d*_33_ value even when bent to a radius of about 10 mm, indicating good piezoelectric performance and mechanical flexibility. The mold used in the freeze-casting method largely determines the structure of the piezoelectric ceramics. By adjusting the mold structure, PZT can be made into various shapes such as rectangular, circular, and square, offering a selective process compared to the dice-fill method. However, this is not the only way to alter the ceramic structure. Sometimes, changing the direction of solvent solidification (cooling conditions) can also produce different porous structures in the ceramics. Fukushima et al. [119] used the freeze-casting method to create 3-1 and 3-2 connected alkali niobate ceramics (NKN)/epoxy composites, as shown in Figure 28. The researchers employed different cooling schemes for the two structures: for the 3-1 structure, they contacted the bottom aluminum foil of the mold with a cooling ethanol bath and insulated the sides with polyurethane to prevent tert-butyl alcohol (TBA) from solidifying from the sides. Conversely, for the 3-2 structure, they insulated the top and bottom of a tubular aluminum mold with the same polyurethane and used cold air in an incubator to cool the TBA from the sides. This resulted in different connectivity structures in the NKN ceramics despite using a similar mold. Therefore, the freeze-casting method offers considerable flexibility in preparing porous ceramics. When developing flexible piezoelectric composites with porous structures, altering the cooling conditions to achieve the desired porous ceramic structure is also a viable approach worth considering.

#### 3.1.7. 3D Printing

3D printing, also known as additive manufacturing (AM), is an emerging technology that rapidly fabricates materials by stacking layers of material. Compared to traditional manufacturing methods, 3D printing technology offers advantages such as high precision, high efficiency, and low cost. In recent years, it has become increasingly active in the research of various piezoelectric composite materials. Grinberg et al. [120] introduced a basic procedure for using 3D printing to create BTO/polyamide composites (BTO/polyamide type); this is shown in Figure 29. The procedure is performed as follows: Firstly, a solution casting method is used to dissolve thermoplastic polyamide 11 (PA11) in dimethylacetamide at high temperatures. Pre-measured BTO powder is then added and the mixture is ultrasonically treated to ensure uniform dispersion of the particles in the solution. Secondly, the solution is filtered and washed with deionized water, then placed in a high-temperature environment to evaporate the solvent. Thirdly, the resulting thin film material is hot-pressed to gradually melt the composite material under high-temperature conditions. Fourthly, the molten polymer is extruded through a low-temperature mold and cut to the required length before being fed into the 3D printer. Finally, the printing parameters are adjusted to ensure the composite material is evenly deposited, completing the 3D printing of the composite material. Figure 30a shows a photograph of the processed BTO/polyamide-type sample. The preparation of pre-assembled materials is quite rigorous before 3D printing, as it affects the printing results and material properties of the composite. Grinberg et al. emphasized that there is a trade-off between the mechanical flexibility and the electromechanical response of the composite material. A lower volume fraction of piezoelectric particles results in better flexibility of the composite, while a higher volume fraction can improve its electromechanical response. Therefore, reasonably adjusting the proportion of each component in the flexible piezoelectric composite material is key to influencing its performance.

Additionally, one of the major highlights of 3D printing technology is its ability to manufacture extremely complex structures without the need for molds, breaking the limitations of traditional piezoelectric composite material processing. This capability is particularly beneficial for the future production of complex-structured flexible devices. Zeng et al. [121] used 3D printing technology to create a honeycomb-like BTO/epoxy composite material, as shown in Figure 30b. This material is classified as a 3-1 type piezoelectric composite based on its connectivity. The BTO ceramic forms a uniformly arranged honeycomb framework with three-dimensional connectivity, while the epoxy resin uniformly fills the BTO ceramic pores in a one-dimensional arrangement, enhancing the material’s stability. Sebastian et al. [122] employed 3D printing technology to fabricate an interconnecting PZT scaffold, which was then combined with epoxy resin to produce a 3-3 type piezoelectric composite material. The Scanning Electron Microscopy (SEM) image of the PZT scaffold is shown in Figure 30c. Experiments revealed that by simply changing the infill parameters during printing, the scaffold structure could be customized to have a wide range of PZT volume fractions. Additionally, the relative dielectric constant, polarization, and piezoelectric charge coefficient of the composite material could be tailored over a broad range. Li et al. [123] developed a layered ferroelectric metamaterial using low-pressure-assisted 3D printing technology. The ferroelectric layer was composed of high Curie temperature Li-KNN (lead-free potassium sodium niobate ceramics) material combined with highly flexible polyvinylidene fluoride-hexafluoropropylene (PVDF-HFP) material. The electrode layer was made from high-conductivity carbon black (CB) filler and highly biocompatible polylactic acid (PLA). The ferroelectric and electrode layers were alternately arranged in the thickness direction, forming the staggered periodic structure shown in Figure 30d. Experimental data indicated that this metamaterial exhibits a high longitudinal piezoelectric strain coefficient (*d*_33_ > 150 pC/N) and superior fracture toughness (~5.5 MPa·m^1/2^), more than three times higher than traditional PZT and BTO ceramics. This makes it valuable for developing energy harvesters, self-powered sensors, and therapeutic electrical stimulators. The current literature suggests that most 3D printing technologies are predominantly used for processing small-sized, highly stable, and rigid piezoelectric composite materials. However, this approach is highly valuable for developing new-generation flexible piezoelectric composites (especially 0-3 type) and their sensors.

In addition to the above seven mainstream preparation methods, there are also methods such as the alignment-casting method and dielectric method [15], which were important schemes used in the early development of piezoelectric composite materials. However, due to issues such as high process costs, low consistency of ceramic output, and long processing time, they are rarely used nowadays. The performance and application value of the developed piezoelectric composites vary depending on the preparation methods used. Table 2 provides a summary of piezoelectric composites developed under different preparation conditions, facilitating reader analysis. This summary includes information on piezoelectric properties, mechanical properties, advantages and disadvantages of the preparation processes, and the potential value of the developed composites.

### 3.2. Discussion on Factors Affecting the Piezoelectric Performance of Flexible Composite Materials

During the preparation process, certain factors can lead to unexpected changes in the final performance of flexible piezoelectric composite materials, such as dielectric relaxation and polarization intensity. These influencing factors are often closely related to the piezoelectricity, uniformity, and permeability of the materials, and thus cannot be ignored during the preparation process, especially when developing small-sized composite films, nano-composite fibers, etc.

(1)Dielectric Relaxation Behavior

The dielectric relaxation behavior in flexible piezoelectric composite materials refers specifically to the phenomenon where the polarization response of the material changes over time under the influence of an electric field. This property is inherent to dielectric materials. Since the performance of flexible composites is highly dependent on their polarization characteristics, dielectric relaxation can lead to delayed polarization responses, thereby affecting their piezoelectric response speed and efficiency. However, this dielectric relaxation behavior can be optimized under appropriate conditions. Wang et al. [124] investigated the effects of fillers (barium titanate, silicon carbide, and graphite) on the dielectric relaxation behavior of PVDF. The study found that compared to barium titanate ceramics, the conductive properties of silicon carbide and graphite significantly suppressed the dielectric relaxation of PVDF, especially when the content exceeded 60 wt%. This is because the addition of conductive materials not only promotes the orientation of PVDF dipoles but also increases the dielectric loss of the material. Moreover, the dielectric relaxation behavior of composites is temperature-dependent. As the temperature increases, the relaxation peak of barium titanate shifts to a higher frequency region, and the higher the ceramic doping content, the more significant the suppression of the relaxation peak. For semiconductive silicon carbide, the higher the temperature, the more pronounced the relaxation peak; for conductors, although the relaxation peak of PVDF is also suppressed, new relaxation peaks appear, becoming more evident at higher temperatures, as shown in Figure 31. Therefore, the selection of fillers and temperature control can both optimize the dielectric relaxation behavior of flexible piezoelectric composite materials.

(2)Polarization

Polarization plays a crucial role in the performance of piezoelectric materials. It refers to the process in which the electric dipoles within a piezoelectric material align along the direction of an external electric field, which is essential for the material to achieve stable piezoelectric properties. During the preparation of flexible piezoelectric composite materials, the polarization process is influenced by many factors, such as electric field strength, temperature, polarization time, and the choice of constituent phases. In recent years, various polarization methods aimed at enhancing the piezoelectricity of composites have garnered significant attention. Hong et al. [125] used corona polarization to polarize BTO/PLA composite fibers, an effective approach for increasing the output voltage response. The schematic of the corona polarization setup is shown in Figure 32a. Tests indicated that BTO/PLA fiber sensors subjected to corona polarization exhibited higher V_p-p_ values under different normal forces, as shown in Figure 32b. Yang et al. [126] reported that the stepwise pulse polarization (PP) method could increase the *d*_33_ value of a modified PZT piezoelectric ceramic to 196 pC/N, which is a 54% improvement compared to the direct current polarization method, significantly enhancing the piezoelectricity of the ceramic material. In summary, these methods are valuable references for developing highly piezoelectric flexible composite materials. Effective polarization can orderly arrange electric dipoles, thereby improving the piezoelectric performance of the materials, resulting in greater electrical output under mechanical stress or greater mechanical deformation under an electric field.

On the other hand, the percolation of nanoparticle doping is also necessary to discuss, as it affects the polarization behavior of composite materials. Here, the percolation threshold can be used to characterize the material’s percolation. Tian et al. [127] suggested that the doping of conductive fillers significantly impacts the induced polarization behavior. Figure 33 shows the curves of induced polarization and remanent polarization for MXene/PVDF composites at different MXene doping levels. It is evident from Figure 33 that the induced polarization exhibits nonlinear changes under varying doping levels. Particularly, when the MXene content approaches the critical percolation threshold of the composite material at 15%, electrons within the composite material are more likely to migrate and form interfacial polarization under the polarization field, leading to a sharp increase in induced polarization.

(3)Determination of Filler Content

Doping piezoelectric or conductive materials can significantly enhance the performance of flexible piezoelectric composites, not only in terms of piezoelectricity but also in mechanical stability and temperature adaptability. However, the doping content of these fillers should not be excessively high. Tian et al. [127] verified the impact of MXene doping levels on the piezoelectric properties of MXene/PVDF composites, as shown in Figure 34. Figure 34a illustrates the relationship between different MXene content and the composite’s *d*_33_ values (red represents the composite *d*_33_, blue represents the *d*_33_ in the polymer matrix, and green represents the interfacial polarization-coupled *d*_33_). As the MXene content increases, the composite’s *d*_33_ initially rises to 60 pC/N and then drops back to near its initial value, with the highest *d*_33_ value achieved at a 15% doping level. A moderate amount of conductive material facilitates electron migration in the polarization field, enhancing piezoelectricity. However, when the doping content exceeds the threshold, the composite forms a continuous conductive network, providing a percolation path for the relaxation of the polarized charges, thus suppressing the piezoelectricity of the composite. This principle is also explained in Figure 34b. Figure 34b shows the output voltage response of MXene/PVDF composites under 25 N force at different MXene doping levels. It can be seen that the MXene/PVDF composite achieves the highest output at a 15% doping level, and the peak voltage response significantly decreases when the MXene doping level continues to increase. Therefore, the doping of conductive materials is closely related to the critical percolation threshold.

Compared to conductive fillers, determining the content of piezoelectric fillers is relatively complex. Based on whether the polymer material itself exhibits piezoelectricity, the determination of fillers can be divided into two categories. The first category is PVDF-based composites. For these composites, doping with high piezoelectric particles should consider the content of the polar electroactive β-phase. Thirumalasetty et al. [128] tested the output voltage of lead-free barium zirconate titanate (BZT)/PVDF-HFP composites with four different doping levels (0, 8 wt%, 16 wt%, 24 wt%), as shown in Figure 35a. The composite achieved a higher output voltage peak at a 16 wt% BZT ceramic doping level, resulting in higher output power. When the doping level increased to 24 wt%, the output voltage of the BZT/PVDF-HFP composite significantly decreased. They analyzed that a certain amount of ceramic doping facilitates the interaction between the negative charges carried by the ceramics and the -CH_2_ groups in PVDF, enhancing the β-phase content within the composite. However, excessive ceramic doping inhibits β-phase formation, resulting in a loss of piezoelectricity. Verma et al. [129] doped PVDF polymers with KNN ceramics to create KNN/PVDF piezoelectric composites. Similar patterns were observed in their research. Figure 35b illustrates the relationship between different KNN doping levels and the open-circuit voltage of the composites. It is evident that when the doping level exceeds 15 wt%, the piezoelectricity of the composite is significantly weakened. Therefore, for polymers with inherent piezoelectricity, higher piezoelectricity is not achieved by simply increasing the number of piezoelectric particles but by focusing on the synthesis of the β-phase after doping. The second category is non-PVDF-based composites. Since these polymers do not possess piezoelectricity, the piezoelectricity of the composites depends on the piezoelectricity of the fillers. Thus, the larger the piezoelectric particle size and the higher the doping level, the stronger the piezoelectricity of the composite; this corresponds to higher output voltage and power values, as shown in Figure 36 [74]. However, increased piezoelectric ceramic content also leads to greater hardness and higher acoustic impedance of the composites, requiring a trade-off based on the application requirements.

(4)Agglomeration

In piezoelectric materials, the phenomenon of agglomeration refers to the tendency of piezoelectric particles to cluster together rather than being evenly distributed within the material. This leads to an uneven distribution of the electric field within the material, causing some piezoelectric particles to inadequately contribute to the overall charge response [130]. Consequently, this adversely affects the piezoelectric performance of the material. Moreover, during the polarization process, agglomeration may result in non-uniform polarization, further degrading the material’s performance and efficiency. In summary, agglomeration negatively impacts both the piezoelectric properties and polarization of piezoelectric materials and should be minimized to ensure the stability and optimization of material performance. Silakaew et al. [131] have suggested that the agglomeration effect caused by the incorporation of ceramic particles into a polymer matrix can significantly degrade the electrical and mechanical properties of the composite material, as illustrated in Figure 37. This is attributed to the anomalous low-frequency dielectric constant behavior induced by the dipolar polarization and interfacial polarization between BTO ceramics and PVDF. Therefore, during the preparation process, using appropriate solvents and ultrasonic treatments to control the particle size and distribution of the ceramic particles can help mitigate agglomeration.

## 4. Application of Flexible Piezoelectric Composite Materials

### 4.1. Underwater Detection

Hydrophones are important devices for sensing underwater sound information. With the gradual deployment of UUVs and micro-robots underwater, hydrophone development is also gradually transitioning towards miniaturization, lightweight, high precision, and other directions [132,133,134]. In response to these underwater detection needs, the utilization of flexible piezoelectric composite materials will provide valuable design ideas for developing hydrophones with high electromechanical properties and conformability. In recent years, there have been relatively few reports on the underwater detection applications of flexible piezoelectric composite materials in the literature, suggesting that there is considerable research potential in this field.

Wang et al. [135] achieved integrated molding of transducer arrays using semi-flexible piezoelectric composite materials, as shown in Figure 38a. Each transducer element is composed alternately of piezoelectric ceramics and epoxy resin, giving the elements a columnar compressive resistance. Flexible rubber is infused between each array element to facilitate the vibration of elements and attenuate lateral coupling between elements. Additionally, influenced by improvements in flexible polymers, the flexible piezoelectric composite material has the ability for unidirectional bending, which also has practical value in developing curved transducers. Inspired by electronic sensing skin, Hao et al. [86] designed a patch-type flexible transducer, as shown in Figure 38b. This transducer is made of 1-3 type piezoelectric composite material that can undergo three-dimensional deformation, hence possessing a certain level of conformability, making it suitable for meeting the surface shell devices in the UUV series. In terms of performance, this transducer meets surface deformations ranging from 100 mm to 200 mm, with a reception response reaching −186.1dB, thereby possessing application advantages of a broad detection range and strong sensing capability. However, due to the adoption of an all-flexible design in composite materials, the flexible composite material may undergo mechanical deformation [13], potentially making this flexible transducer suitable for underwater application scenarios with low hydrostatic pressure or weak mechanical strength. Nevertheless, this issue of mechanical deformation can sometimes be addressed by enhancing the compressive resistance of the piezoelectric units. As early as 2010, Wang et al. [136] proposed a multi-element composite array composed of piezoelectric ceramics, epoxy resin, and cork rubber, as shown in Figure 38c. The elements are integrally formed by a 1-3 PZT/epoxy structure, and the ceramic base is retained in the composite material structure, ensuring good stability and impact resistance for each element. Here, cork rubber is mainly used to strengthen the mechanical vibration of the piezoelectric columns and to alleviate the lateral effects between the elements, facilitating a better concentration of thickness vibration energy. Although this composite structure resolves the issue of mechanical deformation, the ceramic base restricts the bending capability of this piezoelectric composite material and introduces higher lateral coupling effects. In this design, the balanced use of rigid and flexible polymers provides a balance between the compressive resistance of the piezoelectric units and the overall flexibility of the composite material. Similarly, in the 1-3 improved piezoelectric composite array proposed by Qin et al. [91], the original ceramic base was removed (Figure 11a), and silicone rubber was used as the connecting material for the elements, which greatly reduced lateral coupling, leading to a basic increase in the electromechanical coupling coefficient of the elements to around 0.6. Due to the enhancement of the thickness mode, the overall bandwidth of the composite array is also correspondingly extended. Increasing the content of silicone rubber will result in enhanced conformability of this composite array. Thus, the balanced utilization of rigid and flexible polymers not only effectively improves the mechanical deformation problem of flexible piezoelectric composite materials but also enhances their performance, providing a new approach for the development of flexible transducers capable of withstanding high hydrostatic pressure and exhibiting strong impact resistance in the future.

### 4.2. Wearable Sensor

The combination of flexible piezoelectric composite materials with the functions of medical devices enables the development of wearable sensors adapted to human detection needs [137,138,139]. Compared to traditional large-scale medical devices, wearable sensors offer not only convenience in carrying but also greater flexibility and adaptability. Through the signals of wearable sensors, people can also gain real-time control over physiological information, which has been the main reason for the rapid rise and development of this technology in recent years.

Tian et al. [140] proposed a wearable sensor for continuous monitoring of blood pressure and cardiac function, as depicted in Figure 39. The sensor is composed of a non-uniform layered structure of MXene/boron nitride (BN)/P(VDF-TrFE) and was constructed using controlled electrospinning combined with hot pressing technology, thereby overcoming the intrinsic limitations of traditional piezoelectric ceramic materials which are hard and brittle, while simultaneously balancing high piezoelectricity and flexibility. Leveraging the interfacial polarization between MXene and the polymer matrix and the suppression of bulk leakage current by BN, the piezoelectric composite material exhibits a piezoelectric charge coefficient of up to 41.67 pC/N and a piezoelectric voltage constant of 342.9 mV/mN. The sensor exhibits high-pressure response sensitivity, endowing the device with the potential for use in wearable cardiovascular monitoring applications. Huang et al. [141] proposed a modified sensor made of Polyacrylonitrile (PAN)/MXene/polydopamine-modified zinc oxide (PDA@ZnO-5) piezoelectric composite material (PMPO), capable of achieving a sensitivity response of 28.56 V/N within a wide linear range. This sensor exhibited excellent mechanical stability and durability during 3000 cycles of loading-unloading, which offers multiple benefits such as sensing subtle pressure changes and monitoring significant body movements. As depicted in Figure 40, the sensor can exhibit different electrical signals in response to subtle pressure variations caused by changes in the volunteer’s facial expressions. Consequently, the integration of flexibility and piezoelectricity in flexible piezoelectric composite materials has ignited the market value of wearable sensors. These sensors can diagnose human physiological information through electrical signals, making them technological products for the next generation of medical diagnostics. However, since some of these devices require long-term wearing by patients to achieve dynamic monitoring effects, the continuous and effective supply of energy to the devices is also a concern for researchers.

For the energy supply issue, most researchers have proposed nanogenerators as a solution, with many of these power supply devices also utilizing flexible piezoelectric composite materials. Wang et al. [142] utilized electrospinning techniques to fabricate a composite nanofiber comprising a blend of a piezoelectric functional layer and a frictional-electric friction layer, as depicted in Figure 41a. In this design, multi-walled carbon nanotubes are doped into another flexible friction layer patterned with PDMS to expand the initial capacitance; the piezoelectric layer is fabricated using the P(VDF-TrFE) polymer, known for its excellent shaping capability. This device can achieve energy output through different mechanisms, with the peak output voltage, power, and power density under the frictional electric mechanism reaching 25 V, 98.56 μW, and 1.98 mW/cm^3^, respectively. Under the piezoelectric working principle, the peak output voltage, power, and power density are 2.5 V, 9.74 μW, and 0.689 mW/cm^3^, respectively, providing continuous effective electrical energy for wearable sensors. Thirumalasetty et al. [128] reported on a flexible nanogenerator based on BZT ceramic fibers, which not only serves as a power supply device for small wearable devices but also provides a development path for environmentally friendly sensing technologies. Figure 41b illustrates the three-dimensional structure of the nanogenerator after exploding. The sensitive core of this device consists of a mixture of PVDF and BZT phases, exhibiting high energy storage efficiency and piezoelectricity. Research indicates that the 16 wt% BZT composite ink exhibits higher electrical activity in the β-phase. Furthermore, under an external force of 3 N, the device achieved an optimized open-circuit voltage and short-circuit current of 7 Vpp and 750 nA, respectively, demonstrating the significant value of environmentally friendly BZT-based materials in flexible electronics. Additionally, some researchers employ PVDF-based composite materials to address the energy supply issue. Huang et al. [61] modified PVDF polymer with BTO material to produce PVDF/BTO composite material, achieving an output voltage of up to ~20.0 V and a sensitivity of ~132.87 mV/kPa. This significant piezoelectricity and sensitivity enable it to adapt to various human wearable positions, as shown in Figure 41c. Energy harvesting experiments have shown that under a pressure of 10 N, the power density can reach ~58.7 mW/m^2^, indicating long-term durability. This finding provides an important development direction for sustainable self-powered sensors and energy harvesting sensors.

Flexible piezoelectric composite materials play various roles in the field of wearable sensors. Simultaneously, the development of flexible piezoelectric composite materials has greatly inspired research on various functionalities of wearable sensors and has demonstrated enormous potential in the field of medical detection. As research goes deeper, wearable sensors will evolve towards high mechanical flexibility, high sensitivity, high energy supply, high integration, and comfort. Compared to bulky and single-function detection methods, this technology undoubtedly represents a significant advance in medical detection.

### 4.3. Flexible Electronic Skin

Flexible electronic skin has emerged as a hot research topic in recent years, providing significant value in the development of alternative prosthetic sensing, automatic control, and human-machine interface [143]. The sensing materials of electronic skin are typically made of thin, soft, shape-adaptable, and high-sensitivity responsive thin film materials [144], which endow robot systems with strong sensing capabilities. Given these characteristics, flexible piezoelectric composite materials also possess significant application value in this field.

Khanbareh et al. [145] reported on a flexible composite material electronic skin composed of porous polyurethane (PU) and PZT composite, which can be used for information interaction of soft robots, as shown in Figure 42. The uniform spherical micro-pores of polyurethane polymer aim to enhance the piezoelectric voltage constant by reducing the dielectric constant, thereby significantly improving the weak signal perception ability of the electronic skin; while the PZT particles distributed in the polymer matrix endow the material with strong piezoelectricity. Experimental results show that the *g*_33_ of this electronic skin reaches up to 165 mV m/N, about 5 times higher than that achieved by PZT bulk materials, which is beneficial for the development of highly sensitive tactile sensors. Liu et al. [146] designed a large-scale tactile array of electronic skin, as shown in Figure 43. Compared to traditional sandwich-structured large sensors, this array-type electronic skin is expected to have a thinner thickness, simpler manufacturing process, and greater stretchability. In this design, the flexible composite material unit is composed of PZT particles embedded in a PDMS matrix composite, with good uniformity and passive driving mode, enabling high-resolution tactile sensing, simplifying the data collection process, and reducing manufacturing costs.

As a source of information perception, electronic skin plays a crucial role in robotic systems. The development of electronic skin with better flexibility, stronger perception capability, and greater adaptability undoubtedly contributes significantly to the update and iteration of robotic systems. Nowadays, the advancement of next-generation intelligent sensing robots has begun, and this technology is rapidly advancing into fields such as healthcare, industry, and energy [147,148,149,150]. With the organic integration of various electronic skins and complex algorithms, future intelligent sensing robots will be endowed with greater vitality and capability.

### 4.4. Targeted Therapy

Targeted therapy is an important application of flexible piezoelectric composite materials within the medical field. This technique is commonly combined with ultrasonics, which achieves precise and non-invasive treatment of targeted tissues by controlling the acoustic field of the ultrasound transducer. Targeted therapy is mainly divided into two categories: the first category involves making flexible patches (similar to wearable sensors) and implanting them into the body, achieving precise treatment of local areas within the body through the application of external ultrasound stimulation. Li et al. [151] fabricated a wireless, lead-free, battery-free ferromagnetic implant using a PVDF/ZnO composite material, as shown in Figure 44. During use, this implant needs to be wrapped around the nerves in the targeted area. When the external transducer emits sound waves into the body, the implant can convert vibrations into electric pulses, thereby achieving electrical stimulation in the nerves. Wang et al. [152] utilized electrospinning technology to prepare a nanofiber composite material composed of ZnO@PCL (poly (*ε*-caprolactone))/PVDF, exhibiting various antibacterial, immune-modulating, and osteogenic effects. After implantation of this composite material, significant morphological changes were observed in bone tissue from the sixth week to the twelfth week. Experimental results showed that under ultrasound regulation, this composite material significantly enhanced the proliferation, migration, and osteogenic differentiation of mouse embryo osteoblast precursor cells, thereby expanding the application of flexible piezoelectric composite materials to bone tissue recovery. Chen et al. [153] introduced a pomegranate-like structured PEP@BT (piezoelectric polymer@BTO) flexible nanocomposite material, which serves as a potent wireless neural modulation tool for the effective treatment of Parkinson’s disease. Figure 45 depicts the principle of the improvement of Parkinson’s disease by these nanocomposite particles under ultrasound stimulation. Research indicates that under ultrasound stimulation, this PEP@BT nanocomposite material effectively converts ultrasound into electrical energy, thereby regulating intracellular calcium signals and tyrosine hydroxylase levels, activating neural activity and processes in deep brain tissue, and thus contributing to the regulation of Parkinson’s disease.

In addition, another approach is the development of focused ultrasound transducers. Historically, early research indicates that transducers for medical ultrasound are typically constructed from rigid piezoelectric composite materials [36,37,38,39,40]. These transducers can be classified into two types based on their internal material structure: the first type utilizes planar rigid structures of piezoelectric composite materials. Due to the lack of curvature in the composite material, an acoustic lens is often necessary during fabrication to achieve the desired beam focusing. The other type employs piezoelectric composite materials with inherently curved structures, thereby eliminating the need for an acoustic lens during fabrication. Nevertheless, both types of piezoelectric composite materials are non-flexible, which may require longer production cycles and lower yields. In contrast, flexible piezoelectric composite materials can be easily shaped into curved structures, eliminating the need for an acoustic lens in transducer fabrication and facilitating further optimization of transducer dimensions. The 1-3 PZT/PDMS focusing transducer developed by Kim et al. [84] not only simplifies the traditional structure of acoustic lenses but also has a good acoustic focusing effect. When the hydrophone moves along the *xoy* plane to the point where the sound pressure amplitude is maximum, the actual measured focal length of the sound field is 6.5 mm, which is consistent with the design value (6mm), proving the value of this transducer in focused medicine. For example, the PZT-8/PDMS 1-3 focusing transducer developed by Hou et al. [85] can achieve focusing performance without the use of acoustic lenses, as shown in Figure 46. Within the curvature radius range of 17.5 mm~27.5 mm, the lateral resolution change range of the focal position is 0.79 mm~1.21 mm, and the axial resolution change range is 4.45 mm~14.01 mm. High-resolution dynamic imaging can be achieved in a large spatial range, verifying the ability of this focusing transducer to perform non-contact dynamic focusing imaging on a large spatial range. 

The development of flexible composite materials is of significant importance in expanding medical treatment methods and improving transducer structures. In the field of targeted therapy, flexible piezoelectric composite materials serve as adjunctive therapies, representing a burgeoning modern medical approach in recent years, expediting the transition of traditional large medical equipment towards miniaturization, convenience, and efficiency, with promising prospects in the future. In addition to the above four main application directions, flexible piezoelectric composite materials can also be used in research directions such as motion perception and rehabilitation training [154,155,156,157,158,159]. These designs generally utilize the force-sensitive sensing characteristics of flexible piezoelectric composite materials to evaluate physiological information such as posture and touch through pressure signals, and these applications are generally similar to the principles of wearable sensors and electronic skins introduced earlier.

### 4.5. Ultrasound Diagnostics for Deep Tissue

The ultrasound diagnosis of deep tissues has become a popular research area in recent years, primarily leveraging the strong penetration capability of ultrasound technology. This technique involves placing one or several ultrasonic transducers (arrays) on the surface of the skin or within the internal tissues of a biological organism. By utilizing ultrasound feedback or stimulation, this approach achieves real-time monitoring or therapeutic effects, thus categorizing it under wearable sensors. Typically, these ultrasonic transducers employ a 1-3 type piezoelectric composite structure for their piezoelectric elements (1-3 elements). This configuration allows for high-purity thickness vibration energy of the transducing elements while maintaining low insertion loss. Additionally, these transducers are equipped with flexible printed circuit boards or flexible packaging substrates for positioning or embedding the 1-3 elements that generate ultrasound waves. This facilitates flexible/stretchable functionality, providing numerous important insights for the design of ultrasonic medical products.

Faced with the mismatch issues caused by the large size and rigidity of traditional ultrasound equipment, Wang et al. [160] reported a stretchable ultrasound patch composed of 1-3 elements. The 1-3 elements are used to suppress shear modes and enhance the longitudinal transmission capability of acoustic waves. The electrodes on the top and bottom of the elements are interconnected in a serpentine structure to improve the stability of the elements. Tests have demonstrated that this ultrasound patch exhibits excellent mechanical compliance, with a tensile strain of up to 60%, while also being able to generate ultrasound waves at a depth of 4 cm. Furthermore, this stretchable ultrasound patch can adapt to various positions on the body and can dynamically and accurately monitor deep biological tissues or organs through non-invasive methods, offering new insights for the diagnosis and prediction of cardiovascular diseases in a wearable form. Jiang et al. [161] developed a flexible ultrasonic retinal stimulation piezoelectric array (F-URSP) for providing prosthetic vision to people with acquired blindness. This device integrates a two-dimensional piezoelectric array, rectifiers, and a 32-pixel stimulating electrode array on a single flexible printed circuit board, achieving a high level of integration, as shown in Figure 47. Similar to the previously mentioned work, the F-URSP also employs several 1-3 elements as the ultrasonic array, each composed of lead magnesium niobate-lead titanate (PMN-PT) single crystal with strong piezoelectric properties and epoxy resin with good stability. This configuration not only provides a high electromechanical coupling coefficient (*k*_t_ ~ 0.84) but also ensures a high piezoelectric voltage coefficient (*g*_33_ ~ 40.3 × 10^−3^ V m N^−1^). Experimental results show that when the stimulating electrode array of the F-URSP is matched with the isolated retinal tissue of mice and induced by ultrasound, it can cause a 5% change in the average fluorescence intensity on the retina, verifying the feasibility of ultrasound-induced activation of neural cell activity.

Developing flexible/stretchable ultrasonic transducers (arrays) provides feasible solutions for medical dynamic monitoring and induced therapy, significantly enhancing convenience. Besides external treatments, these ultrasonic transducers (arrays) can also serve as targeted stimulation implants, which enables ultrasound-induced deep tissue repair. In this context, the environmental friendliness or biocompatibility of the functional materials is an important consideration. Wang et al. [162] developed a flexible biphasic ultrasound implant (f-BUI), as shown in Figure 48. The f-BUI contains 1-3 elements with different resonant frequencies, rectifiers, and transistors, all integrated into a small flexible printed circuit board measuring 13.6 mm × 7.3 mm, achieving a high level of integration. This study has two notable highlights: firstly, KNN was used in developing the 1-3 elements, balancing enhanced biocompatibility with piezoelectric properties. Secondly, the researchers reverse-connected two pairs of rectifiers and transistors with 1-3 elements of different resonant frequencies (1 MHz and 3 MHz). By regulating the ultrasound frequency and phase, they achieved bipolar electrical stimulation, reducing charge accumulation and adverse electrochemical reactions on the electrode surface, thereby minimizing tissue damage. Experimental results demonstrated that, when induced by ultrasound, the f-BUI effectively suppressed the overall intensity of abnormal electrocorticography (ECoG) signals in epileptic mice, alleviating the severity of epilepsy and confirming the reliability of the f-BUI implant in treating epilepsy.

## 5. Conclusions and Outlook

This paper reviews the progress of research on flexible piezoelectric composite materials both domestically and internationally; introduces the types and preparation methods of flexible piezoelectric composite materials; and explores their applications in various fields. Flexible piezoelectric composite materials have demonstrated their value in research due to excellent sensitivity, mechanical flexibility, and high adaptability to sensing systems. In contrast with traditional piezoelectric composite materials, the emergence of flexible piezoelectric composite materials will enable the creation of an abundance of structural designs and application scenarios, thus greatly assisting in the development of future intelligent systems. At present, flexible piezoelectric composite materials are undergoing rapid development. Although many outstanding achievements have already been made, there are still aspects that require further optimization:One critical aspect is non-toxicity and environmental sustainability. The enhancement of piezoelectric properties in flexible piezoelectric composite materials often involves doping with high lead content PZT powder, which is a ceramic material known for its high lead content, which poses potential harm to human health and the environment with prolonged use. Besides PZT, improved ZnO materials are occasionally considered for flexible piezoelectric composite materials. However, due to their toxic and polluting nature, these inherent drawbacks may limit the potential applications of such materials. Therefore, advancements in non-toxic and environmentally friendly piezoelectric materials have the potential to further expand the application scope of flexible piezoelectric composite materials. Furthermore, some researchers have integrated lead-free ceramics like BZT into applications involving piezoelectric composite materials, suggesting that flexible piezoelectric composite materials developed using such piezoelectric materials may hold significant economic advantages.Functional integration of sensing structures. The inception of wearable sensors and electronic skin has initiated a new epoch in advancing the design of flexible piezoelectric composite materials towards miniaturization, convenience, and mass accessibility. However, many of these sensors are designed with relatively single functions, making it difficult to address complex integrated problems. For this reason, flexible piezoelectric composite materials may require smaller structural sizes to promote the multi-functional integration of sensors.Further optimization of performance trade-offs is essential. The development of flexible piezoelectric composite materials with high-performance trade-offs is crucial for resolving various trade-offs. For example, in 0-3 type piezoelectric composite materials, it is crucial to consider the balance between material flexibility and piezoelectricity. When the piezoelectric ceramic powder is excessively doped, the material’s flexibility will significantly decrease, while when the doping amount is too small, the material’s piezoelectricity will be greatly compromised. However, piezoelectric composites developed through doping tend to exhibit lower piezoelectricity compared to existing piezoelectric ceramics. Therefore, to meet performance requirements, the development of flexible piezoelectric materials that can simultaneously balance flexibility and high piezoelectricity will become extremely important. On the other hand, multi-component flexible piezoelectric composites, comprising three or more materials, demonstrate great development potential. Since achieving compression resistance and flexibility simultaneously in the same material poses challenges, this multi-component design approach provides support for balancing compression resistance and flexibility. In complex underwater environments, this approach might be the most optimal solution.

## Figures and Tables

**Figure 1 micromachines-15-00982-f001:**
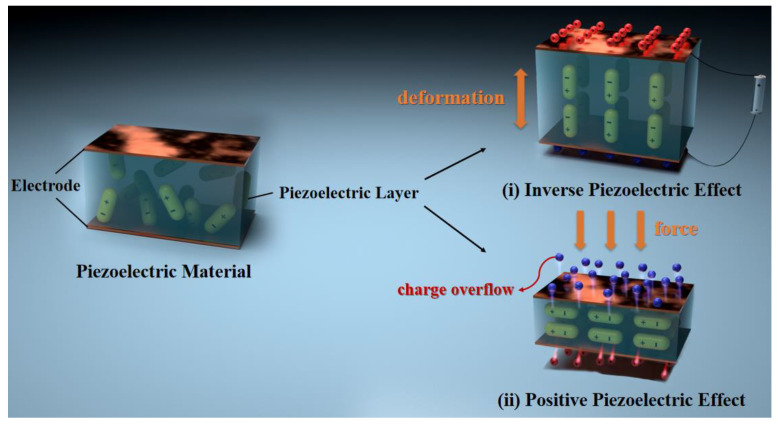
Schematic diagram of the principles of the piezoelectric effect. (**i**) Inverse Piezoelectric Effect. (**ii**) Positive Piezoelectric Effect.

**Figure 2 micromachines-15-00982-f002:**
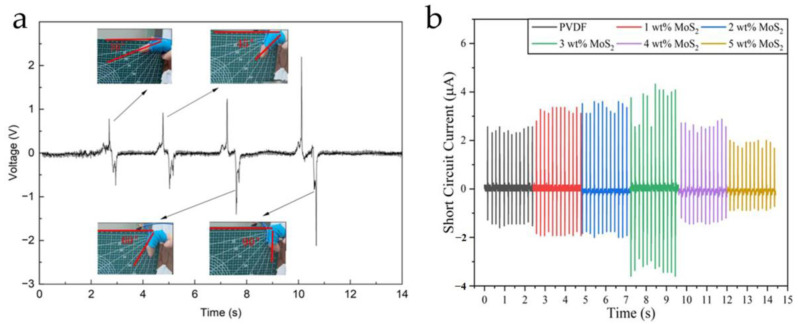
(**a**) Output voltage response of the piezoelectric sensor under different degrees of finger bending [11]. (**b**) The current response curve of the piezoelectric nanogenerator under different MoS_2_ doping contents [12].

**Figure 3 micromachines-15-00982-f003:**
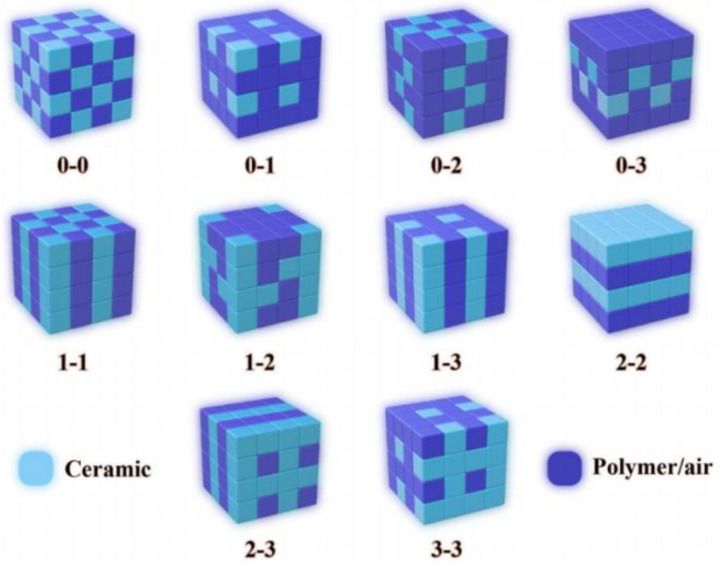
Diagram of ten connected structures of piezoelectric composite materials [22].

**Figure 4 micromachines-15-00982-f004:**
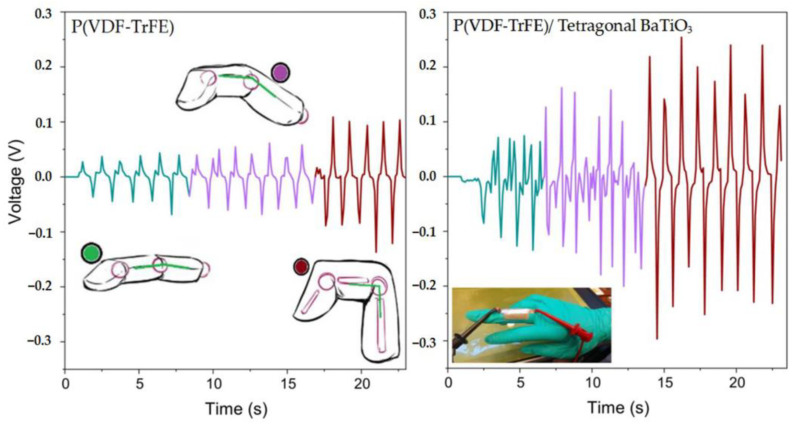
Voltage response of BTO/P(VDF-TrFE) and pure P(VDF-TrFE) materials under *g*_31_ mode [71].

**Figure 5 micromachines-15-00982-f005:**
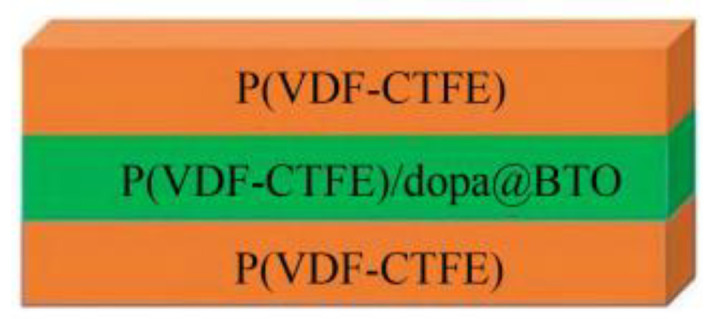
Layered structure of BTO/P(VDF-CTFE) composite material [72].

**Figure 6 micromachines-15-00982-f006:**
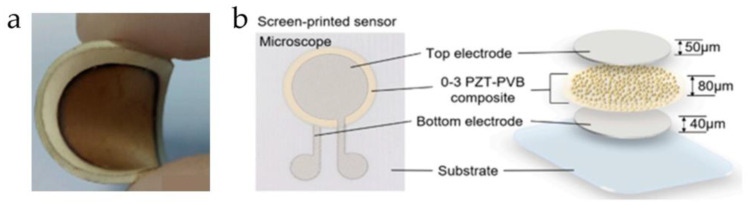
(**a**) Sample of 0-3 PZT/Silicon resin composite material [74]. (**b**) Force-sensitive sensor of 0-3 PZT/PVB type [76].

**Figure 7 micromachines-15-00982-f007:**
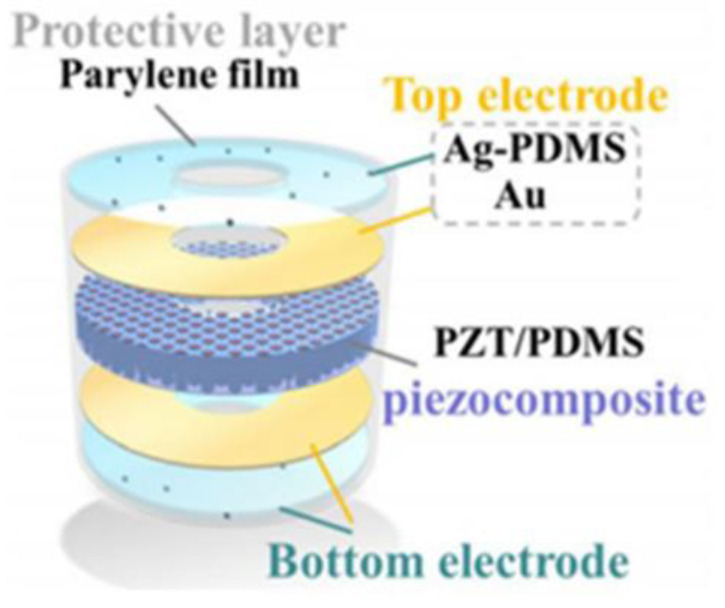
Structure of 1-3 PZT-8/PDMS flexible composite material [85].

**Figure 8 micromachines-15-00982-f008:**
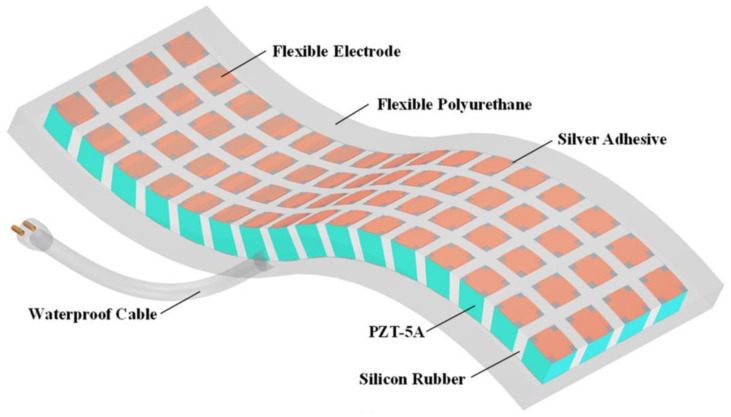
Flexible underwater acoustic transducer with significant deformability [86].

**Figure 9 micromachines-15-00982-f009:**
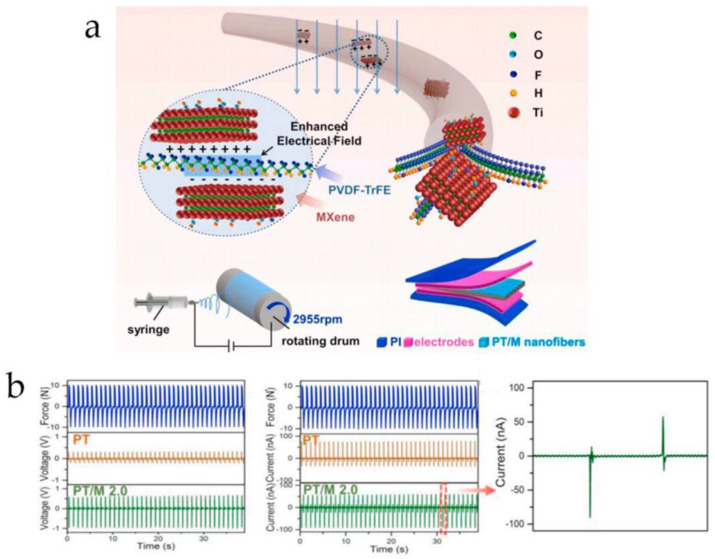
(**a**) Structure of the piezoelectric composite fiber and sensor of XNenes/P(VDF-TrFE) developed by Wang et al. (**b**) Piezoelectric response test results of XNenes/P(VDF-TrFE) compared to pure P(VDF-TrFE) [88].

**Figure 10 micromachines-15-00982-f010:**
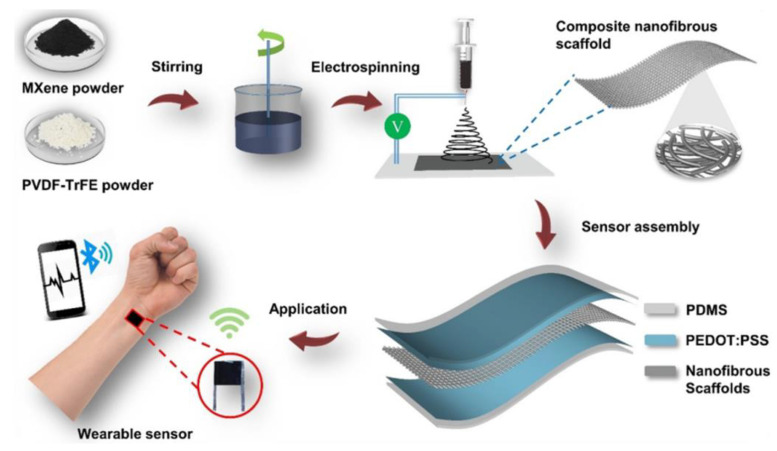
Enhanced composite film of XNenes/P(VDF-TrFE) developed by Sharma et al. [89].

**Figure 11 micromachines-15-00982-f011:**
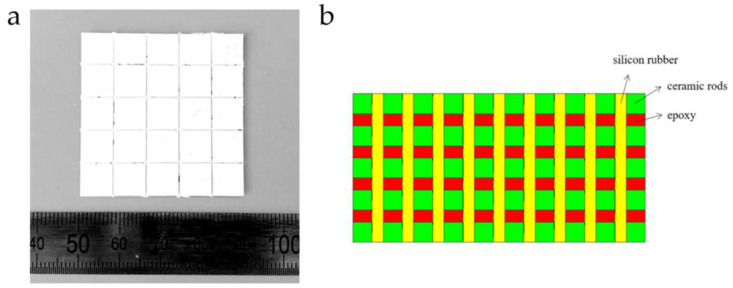
(**a**) Large-scale 1-3 piezoelectric composite array [91]. (**b**) Structure diagram of semi-flexible nested piezoelectric composite material [92].

**Figure 12 micromachines-15-00982-f012:**
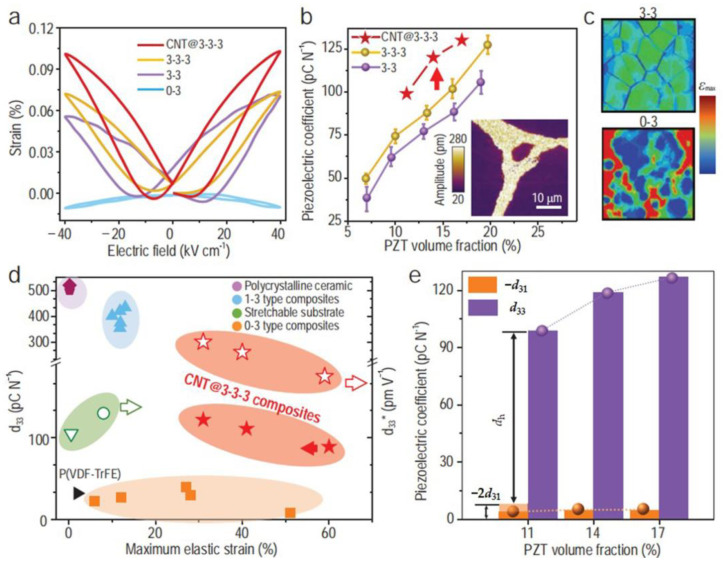
Performance test chart of 3-3-3 piezoelectric composite material [93]. (**a**) The S-E loop of ‘3-3’, ‘3-3-3’, and CNT@‘3-3-3’ composites with 14% PZT volume fraction, and the ‘0-3’ composite with 20% PZT volume fraction. (**b**) The relationship between the piezoelectric charge coefficient and PZT volume fraction for the ‘3-3’, ‘3-3-3’, and CNT@‘3-3-3’ composites. (**c**) Finite element simulation results of maximum strain distribution under uniaxial tension (nominal strain = 5%) for ‘0-3’ and ‘3-3’ composites. (**d**) Comparison of *d*_33_ (solid symbols) and *d*_33_* (hollow symbols) at maximum elastic strain between CNT@‘3-3-3’ composites (red star) and other representative piezoelectric materials. (**e**) The variation of *d*_33_ and *d*_31_ with PZT skeleton volume fraction in CNT@‘3-3-3’ composites. The hydrostatic piezoelectric coefficient *d*_h_ follows the equation *d*_h_ = *d*_33_ + 2*d*_31_, where the *d*_31_ values for both PZT and PZT-based composites are negative.

**Figure 13 micromachines-15-00982-f013:**
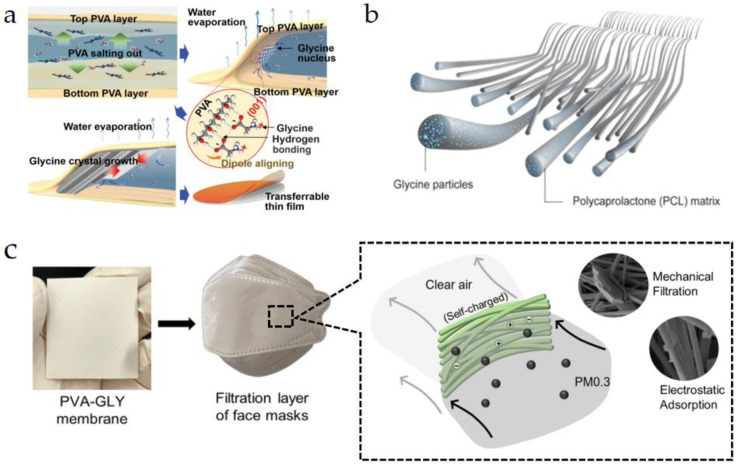
(**a**) Three-layer sandwich-structured PVA-glycine-PVA bio-composite film [94]. (**b**) Glycine-PCL fiber composite material [95]. (**c**) A glycine-PVA composite mask [96].

**Figure 14 micromachines-15-00982-f014:**
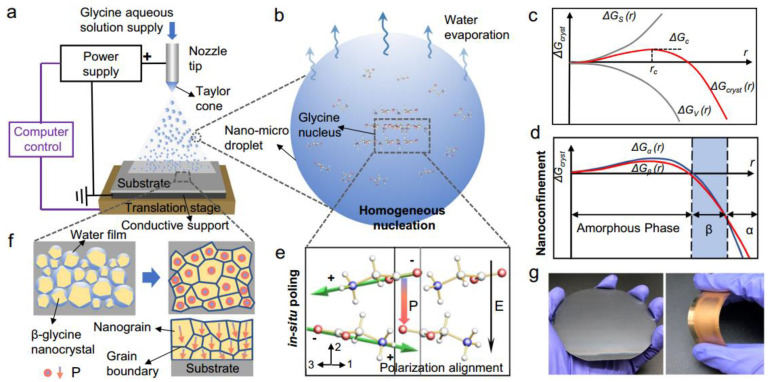
Schematic diagram of the principle for preparing β-glycine piezoelectric nanocrystal films using a coordinated approach of nanoscale confinement and in-situ polarization [98]. (**a**) Diagram of the synthesis principle of a bio-organic thin film printer and β-glycine film. (**b**) Schematic diagram of the nano microdroplets of glycine solution and its crystallization process. (**c**) Free energy (ΔGcryst) curve of crystal nucleation as a function of crystal radius r. (**d**) Size-related free energy distribution diagrams of the two competing nuclei corresponding to α-glycine and β-glycine. (**e**) Schematic diagram of the orientation arrangement of glycine molecules during the homogeneous nucleation process. (**g**) Photos of the thin film on a 4-inch silicon wafer (**left**) and the thin film on a flexible gold-plated polyethylene terephthalate (PET) substrate (**right**). (**f**) Schematic of the film formation process showing the compact nanograins with uniform and consistent polarization orientation (red spot and red arrow in nanograins). The top two images are the surface view of the films, and the bottom image is the cross-sectional view.

**Figure 15 micromachines-15-00982-f015:**
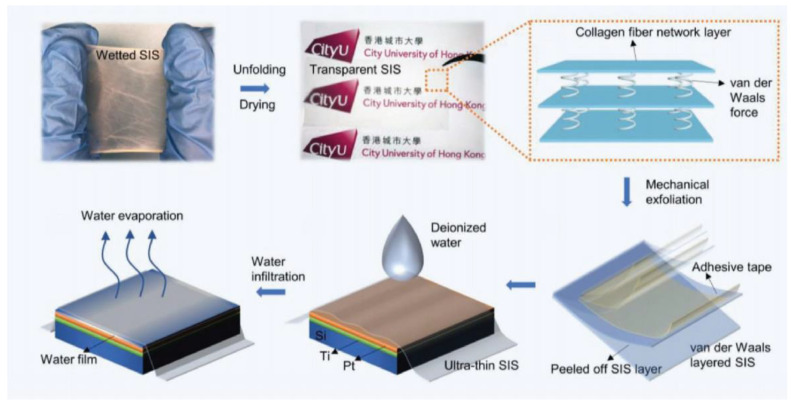
Process flow for preparing SIS thin films and PFM sample preparation [100].

**Figure 16 micromachines-15-00982-f016:**
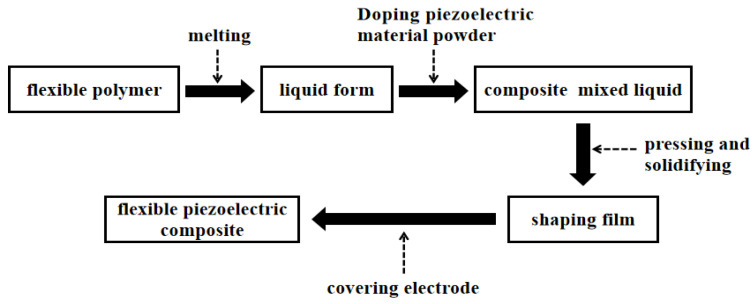
Schematic diagram of preparing flexible piezoelectric composite material film by hot pressing method.

**Figure 17 micromachines-15-00982-f017:**
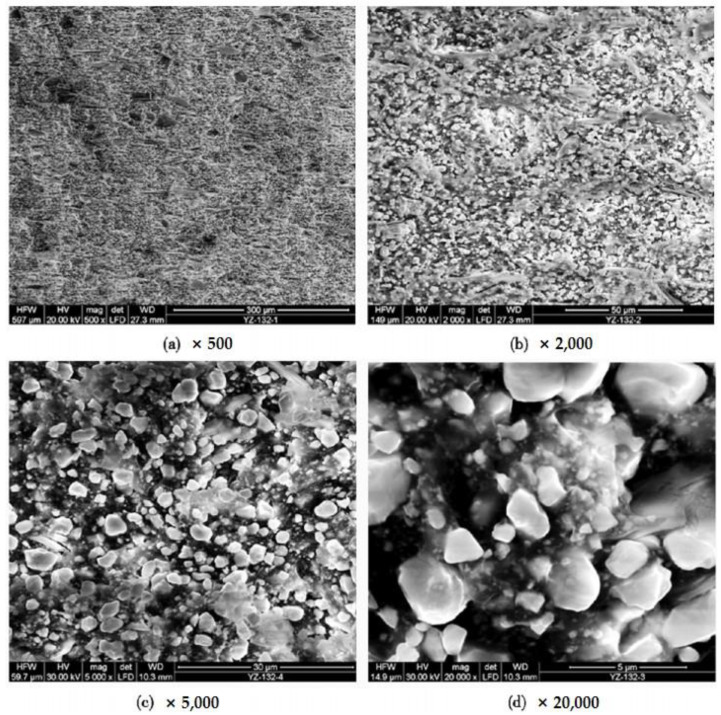
Microscopic images of 0-3 rubber-based damping materials at different magnifications [103].

**Figure 18 micromachines-15-00982-f018:**
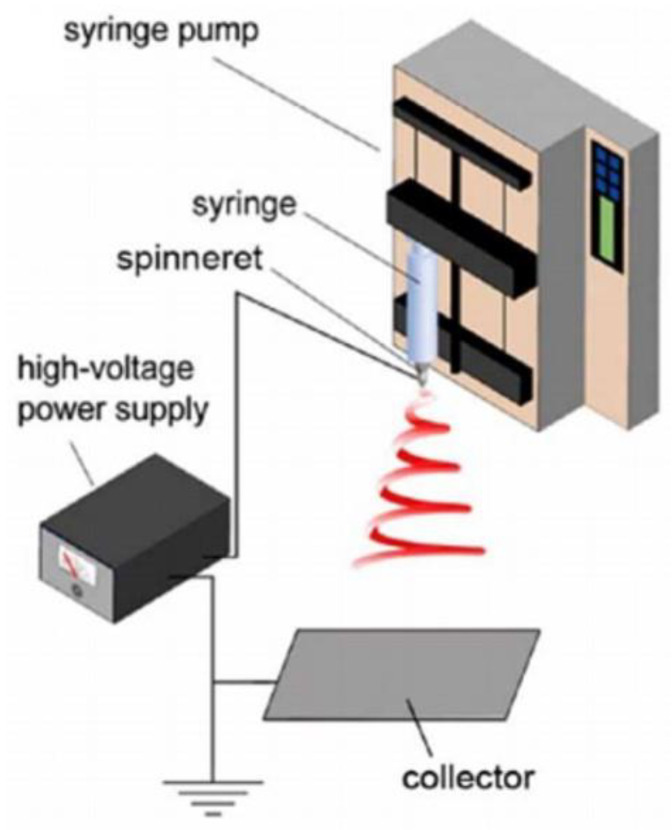
Schematic diagram of electrospinning apparatus [91].

**Figure 19 micromachines-15-00982-f019:**
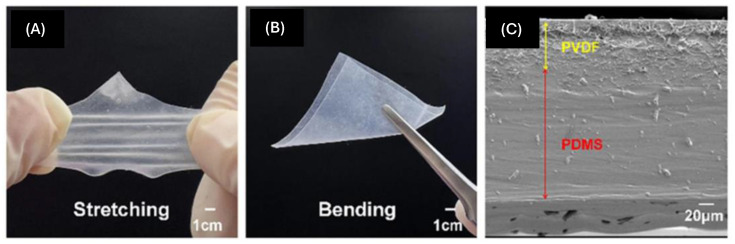
PVDF/PDMS composite film prepared by electrospinning [108]. ((**A**) Picture of PVDF/PDMS composite film under tension. (**B**) Picture of PVDF/PDMS composite film under folding. (**C**) Microscopic images of PVDF/PDMS composite films.).

**Figure 20 micromachines-15-00982-f020:**
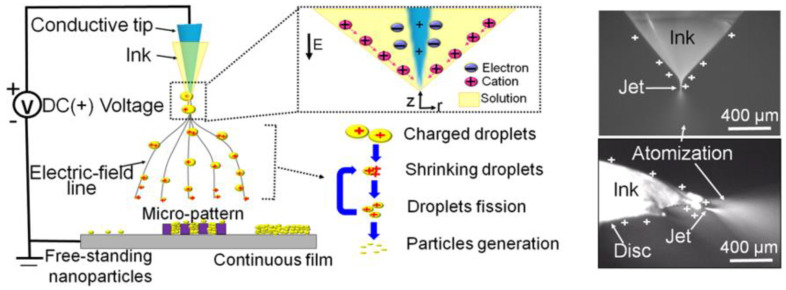
Schematic diagram of the principle of electrospray deposition method [110].

**Figure 21 micromachines-15-00982-f021:**
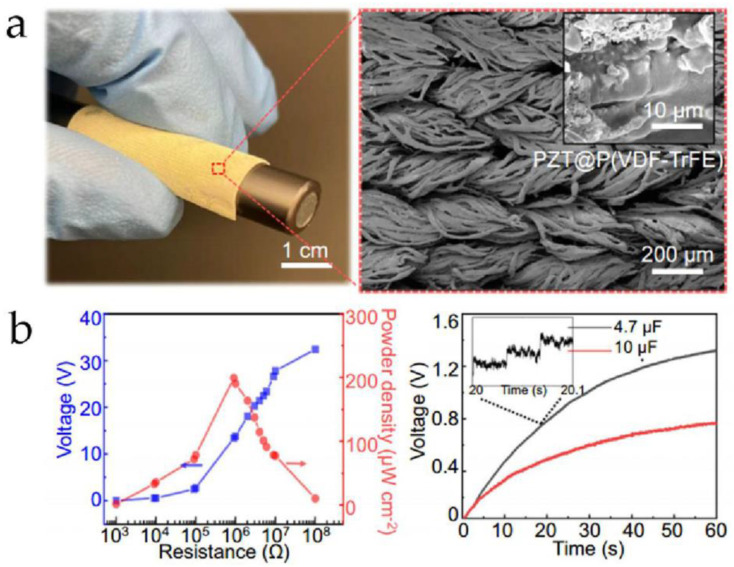
(**a**) Photographs of HIPCT samples prepared by electro-spray deposition and their microstructures. (**b**) Voltage response test curve of HIPCT [110].

**Figure 22 micromachines-15-00982-f022:**
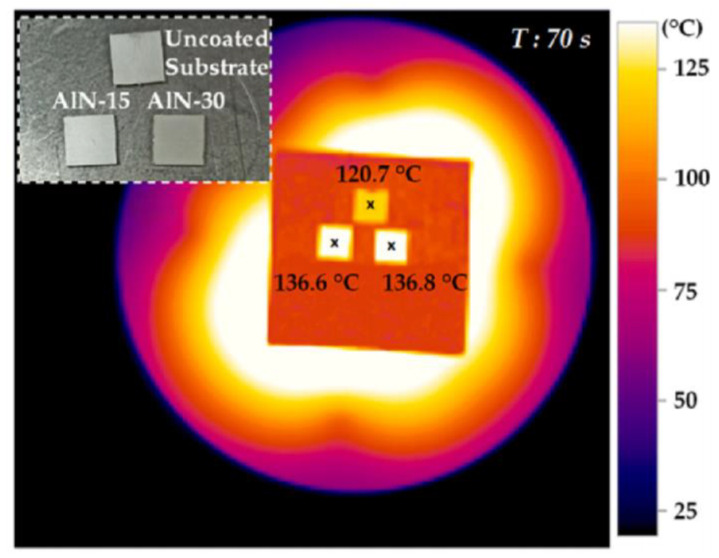
Thermal images of three samples with different AlN-PVA spraying durations (uncoated, sprayed for 15 min, and sprayed for 30 min) heated on graphite foil for 70 s [111].

**Figure 23 micromachines-15-00982-f023:**
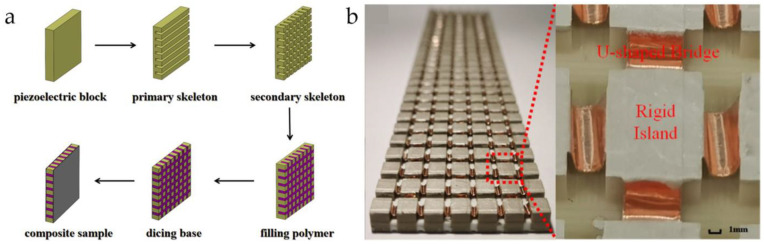
(**a**) Schematic diagram of the dice-fill process; (**b**) Interconnected electrode structure of 1-3 flexible piezoelectric composite materials [87].

**Figure 24 micromachines-15-00982-f024:**
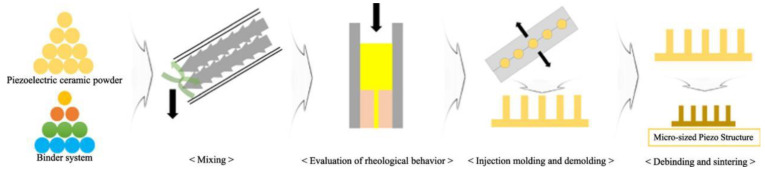
Schematic diagram of the injection-molding process [112].

**Figure 25 micromachines-15-00982-f025:**
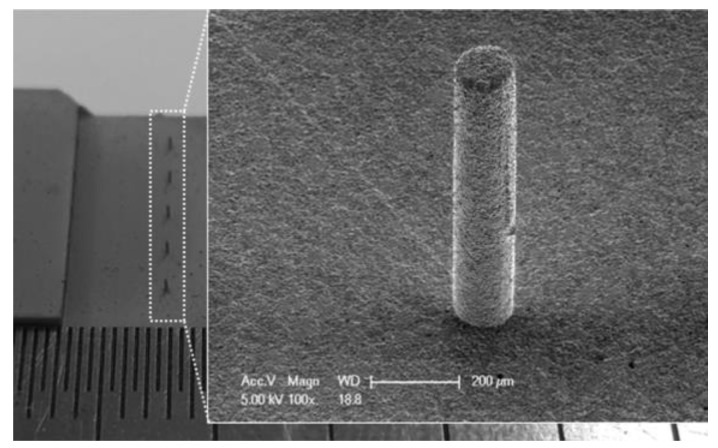
Microscope image of the prepared piezoelectric ceramic micro-pillar structures [112].

**Figure 26 micromachines-15-00982-f026:**
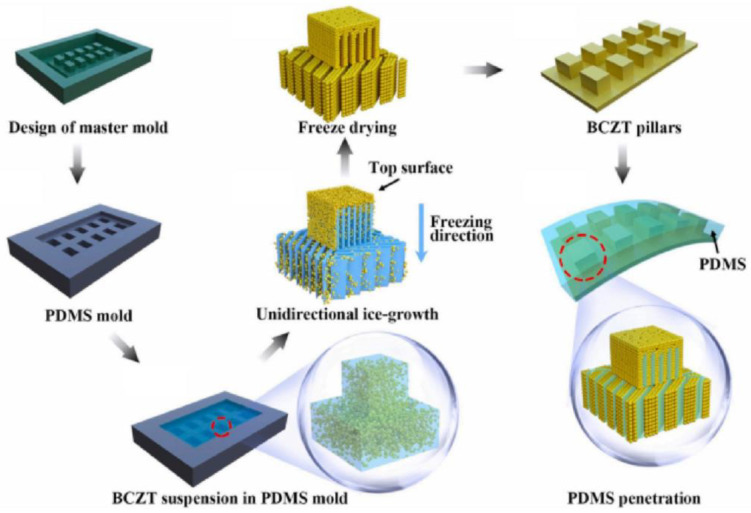
Process flowchart for preparing flexible BCZT/PDMS composites using the freeze-casting method [117].

**Figure 27 micromachines-15-00982-f027:**
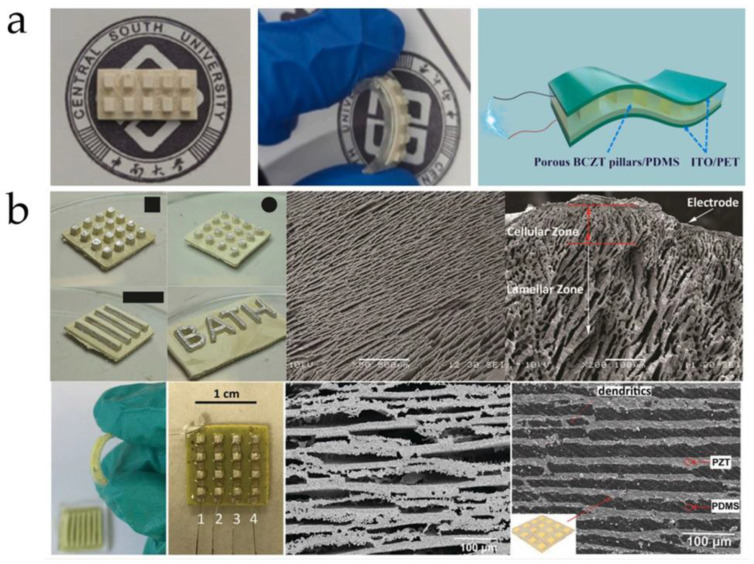
(**a**) Sample image of BCZT/PDMS composite material prepared using the freeze-casting method [117]. (**b**) PZT/PDMS composite material samples with different ceramic structures and their morphology characterization under a microscope [118].

**Figure 28 micromachines-15-00982-f028:**
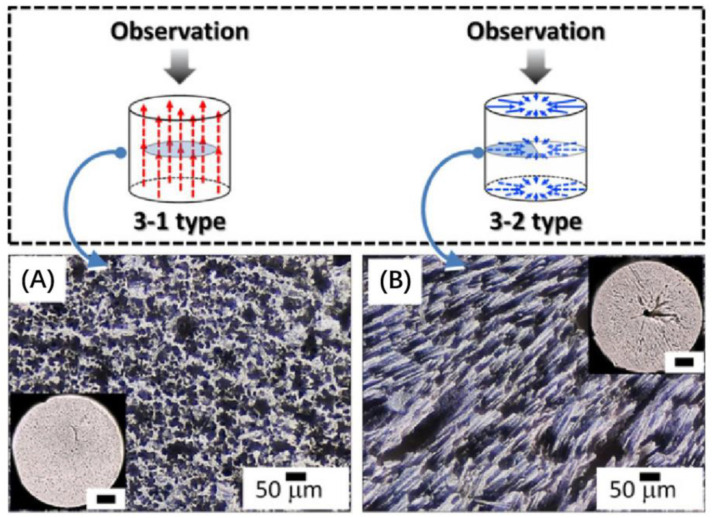
Micrographs of 3-1 type (**A**) and 3-2 type (**B**) NKN/epoxy composites were prepared using the freeze-casting method with different cooling approaches [119].

**Figure 29 micromachines-15-00982-f029:**
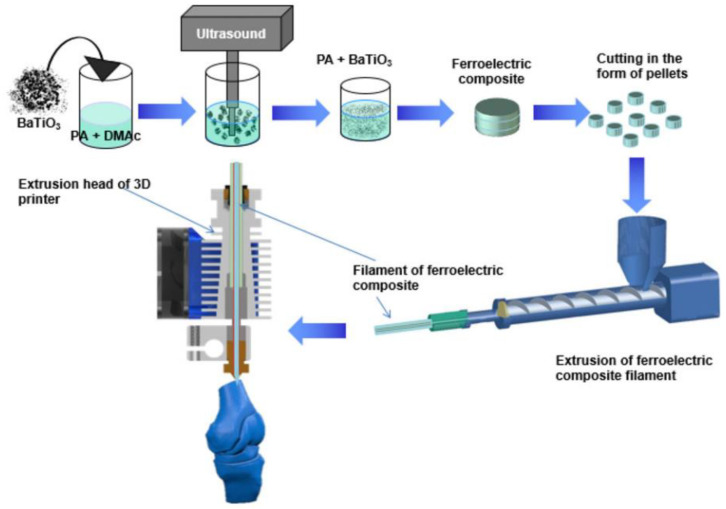
Schematic of BTO/Polyamide Composite Fabrication Using 3D Printing Technology [120].

**Figure 30 micromachines-15-00982-f030:**
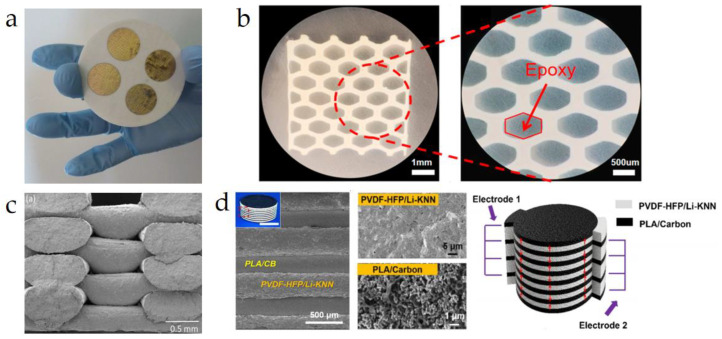
(**a**) BTO/Polyamide composite [120]. (**b**) Honeycomb-structured BTO/Epoxy composite [121]. (**c**) SEM image of PZT scaffold fabricated using 3D printing [122]. (**d**) Layered ferroelectric metamaterial [123].

**Figure 31 micromachines-15-00982-f031:**
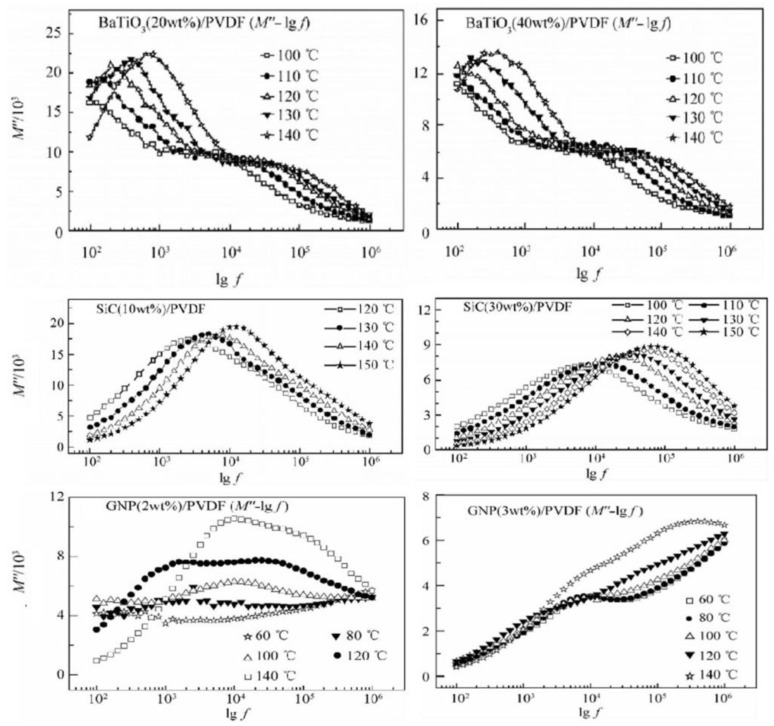
Study on the Dielectric Relaxation of PVDF Composites with Barium Titanate, Silicon Carbide, and Graphite Fillers [124].

**Figure 32 micromachines-15-00982-f032:**
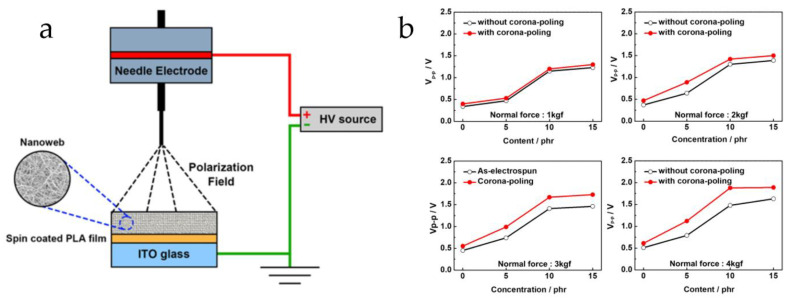
(**a**) Schematic of the corona polarization setup for BTO/PLA composite fibers. (**b**) Voltage response of BTO/PLA fiber sensors with and without corona polarization under different normal forces [125].

**Figure 33 micromachines-15-00982-f033:**
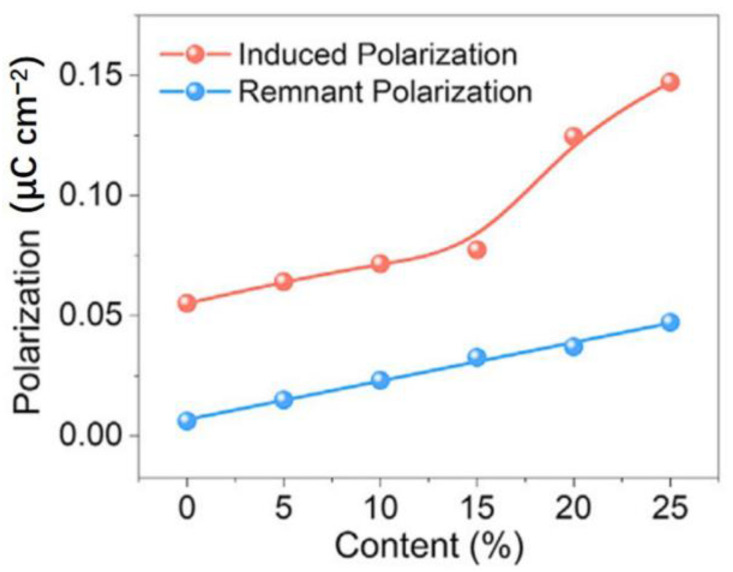
Curves of induced polarization and remanent polarization for MXene/PVDF composites at different MXene doping levels [127].

**Figure 34 micromachines-15-00982-f034:**
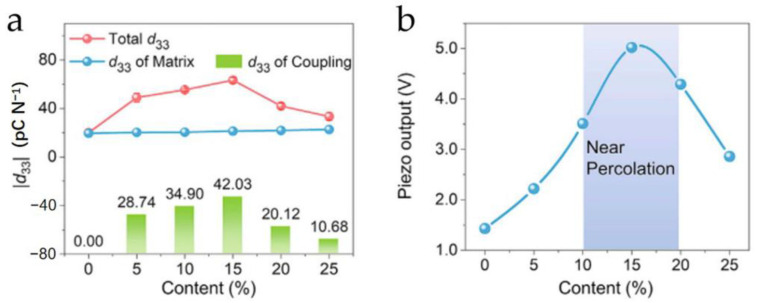
(**a**) Relationship between different MXene content and *d*_33_ of MXene/PVDF composites [127]. (**b**) The output voltage response of MXene/PVDF composites under 25 N force at different MXene doping levels.

**Figure 35 micromachines-15-00982-f035:**
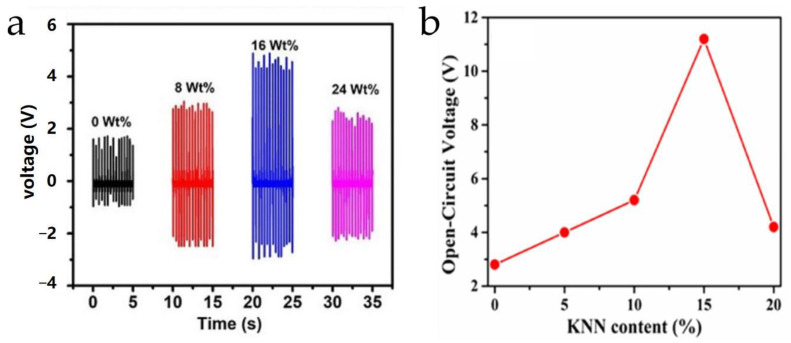
(**a**) Output voltage values of BZT/PVDF-HFP composites with different doping levels [128]. (**b**) Open-circuit voltage values of KNN/PVDF composites at different KNN doping levels [129].

**Figure 36 micromachines-15-00982-f036:**
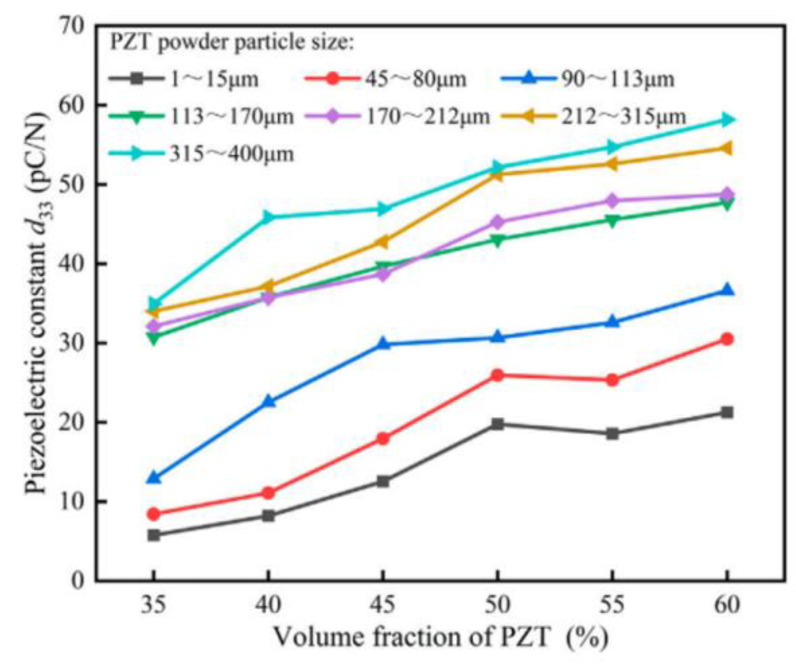
Effect of PZT particle size and volume fraction on *d*_33_ of 0-3 PZT/silicone resin composites [74].

**Figure 37 micromachines-15-00982-f037:**
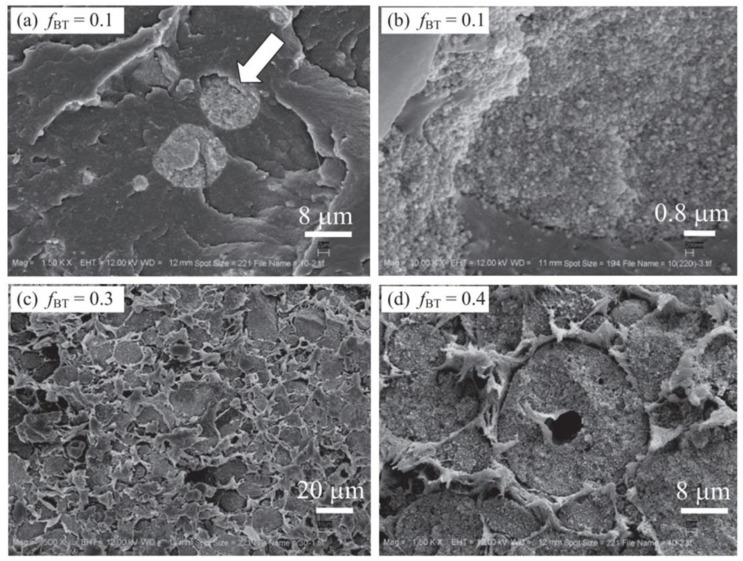
Scanning electron microscope images of the fracture surfaces of BTO/PVDF composites prepared using the dissolution method at different BTO volume fractions [131].

**Figure 38 micromachines-15-00982-f038:**
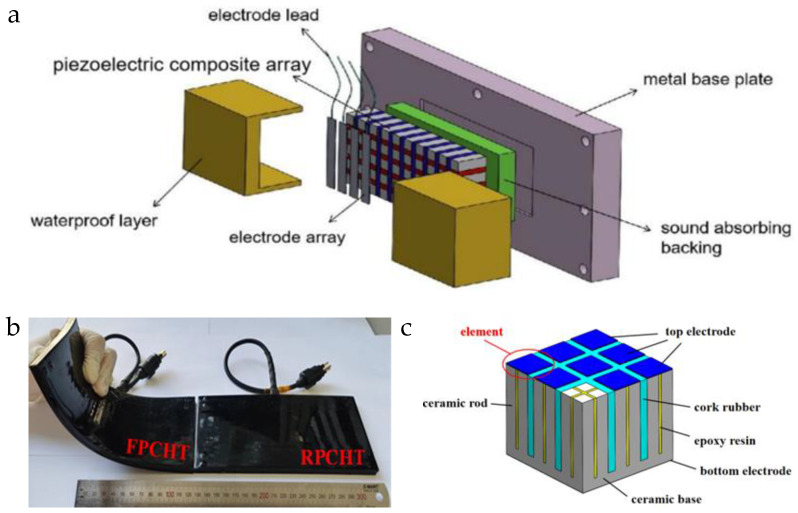
(**a**) Structure of a piezoelectric composite array transducer with semi-flexible characteristics [135]. (**b**) Adhesive-type flexible transducer [86]. (**c**) Multi-element piezoelectric composite array formed by integral molding [136].

**Figure 39 micromachines-15-00982-f039:**
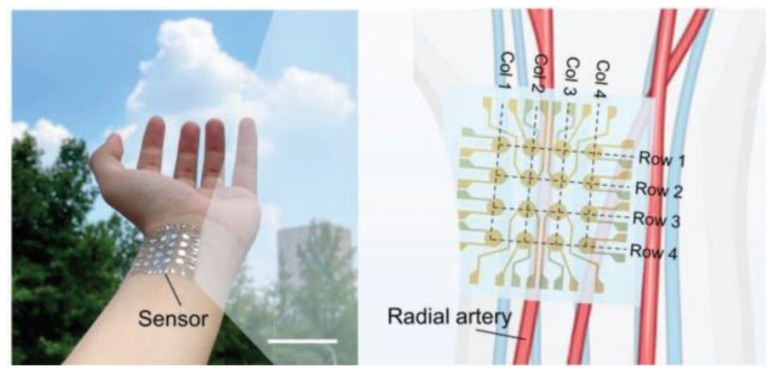
Wearable continuous blood pressure and cardiac function monitoring sensor [140].

**Figure 40 micromachines-15-00982-f040:**
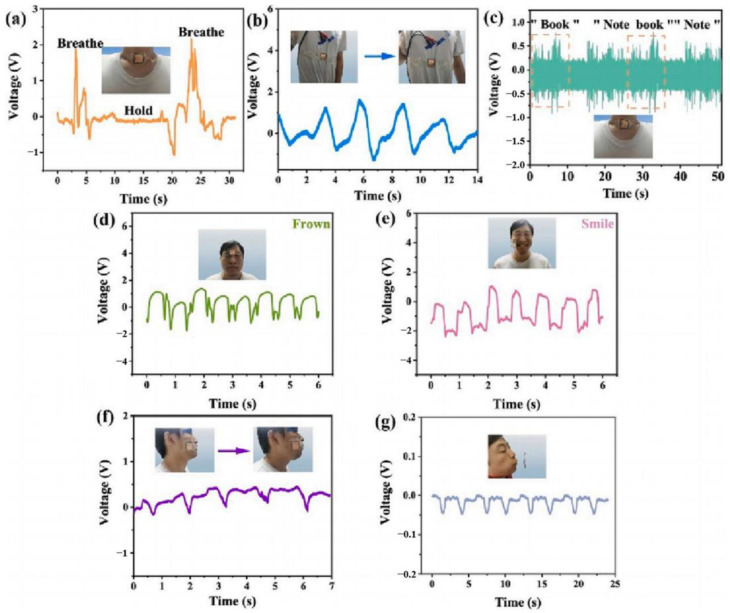
The PMPO sensor is used to monitor subtle body signals: (**a**) breathing, (**b**) abdominal breathing, (**c**) speaking, (**d**) frowning, (**e**) smiling, (**f**) cheek puffing. (**g**) blowing air at a distance of 2 cm from the PMPO piezoelectric sensor [141].

**Figure 41 micromachines-15-00982-f041:**
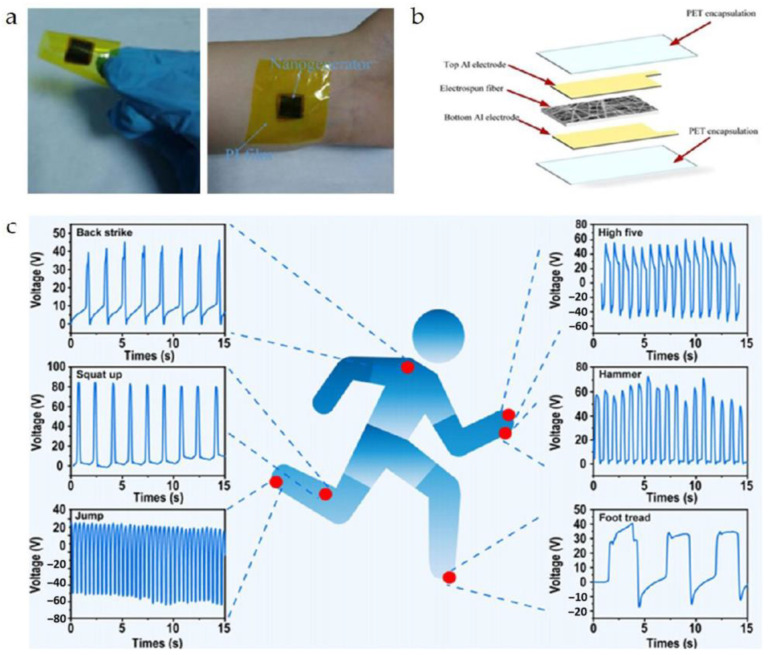
(**a**) Flexible nanogenerator with piezoelectric layer/friction layer [142]; (**b**) Environmentally friendly nanogenerator with PVDF/BZT composite material [128]; (**c**) Voltage signal test results of PVDF/BTO at different wearing positions [61].

**Figure 42 micromachines-15-00982-f042:**
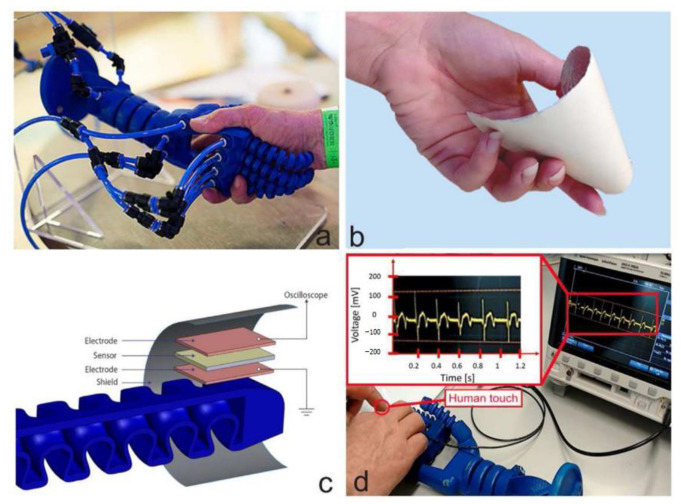
Electronic skin is composed of PZT-PU composite with microporous polyurethane and PZT phase [145]. (**a**) Fully 3D-printed soft robotic hand. (**b**) Large area flexible micro-porous piezoelectric sensor. (**c**) Sensor unit lay-up. (**d**) Recorded signal upon repeatedly touching the sensor.

**Figure 43 micromachines-15-00982-f043:**
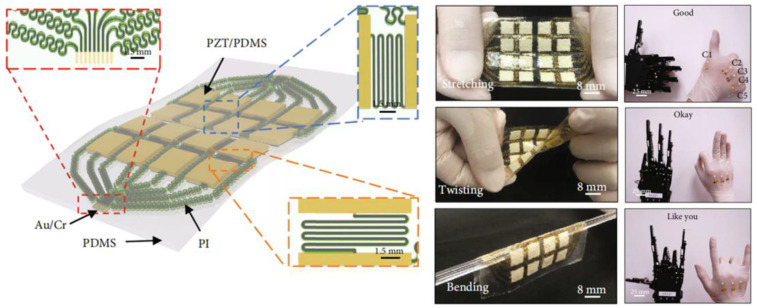
PZT/PDMS electronic skin for gesture recognition [146].

**Figure 44 micromachines-15-00982-f044:**
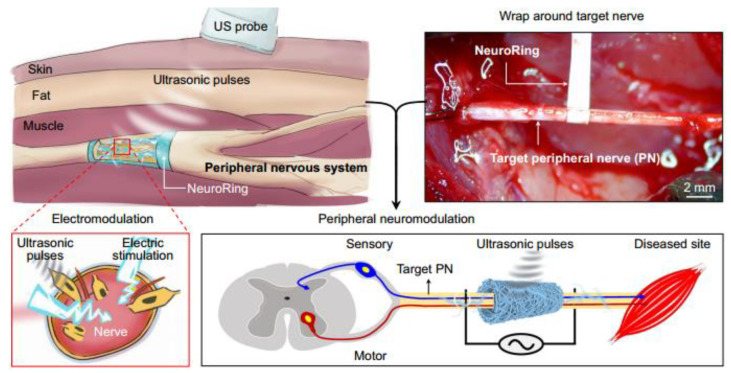
Implantation of PVDF/ZnO flexible composite material and its ultrasound control principle [151].

**Figure 45 micromachines-15-00982-f045:**
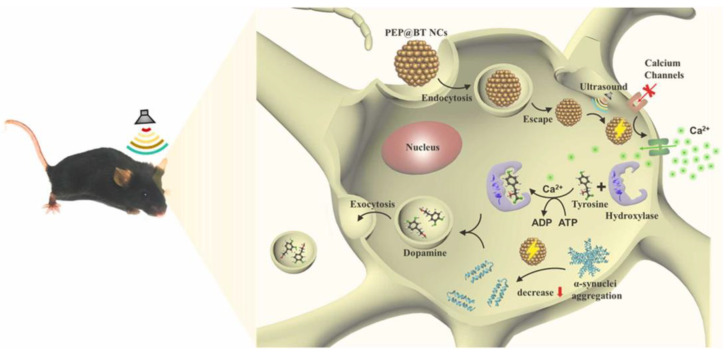
Schematic diagram of PEP@BT nanoparticles promoting recovery of Parkinson’s disease neurons in mice under ultrasound stimulation [153].

**Figure 46 micromachines-15-00982-f046:**
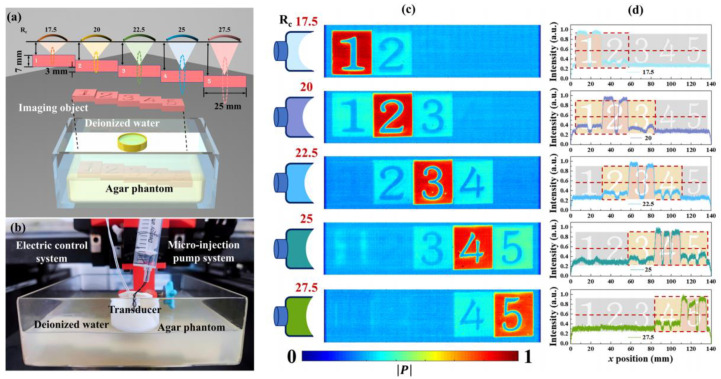
Dynamic acoustic field modulation experiment of focused transducer under different curvatures [85]. (**a**) Schematic diagram of the experimental device includes the size of the digital plate (imaging object) and the principle of dynamic multifocal imaging with the transducer at different curvatures. (**b**) Photogram of the experimental setup. (**c**) Ultrasound imaging of digital plates based on continuous dynamic adjustment method. (**d**) Signal extraction and analysis of ultrasound imaging results.

**Figure 47 micromachines-15-00982-f047:**
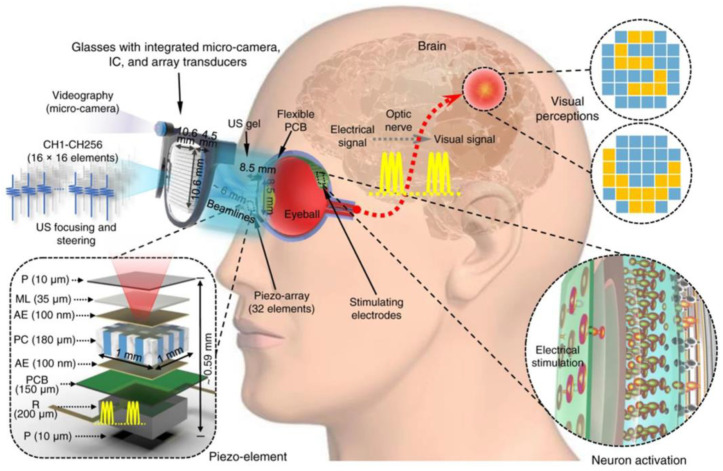
A flexible ultrasonic retinal stimulation piezoelectric array (F-URSP) [161].

**Figure 48 micromachines-15-00982-f048:**
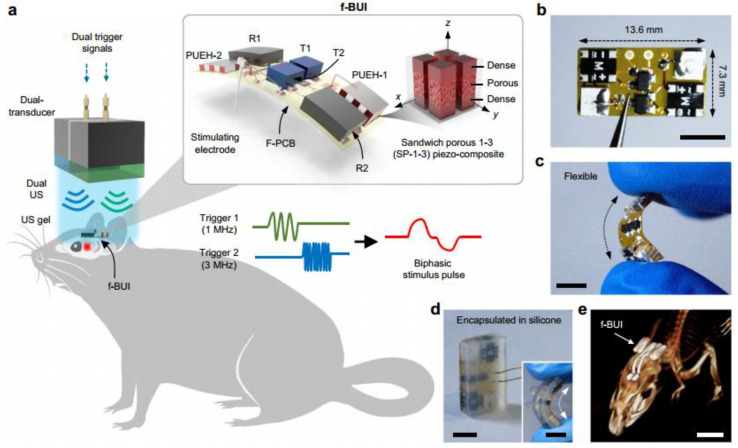
A flexible biphasic ultrasound implant for epilepsy regulation (f-BUI) [162]. (**a**) Schematic diagram of the f-BUI implanted in a rat brain. (**b**) Image of the f-BUI sample before encapsulation. (**c**) f-BUI sample in a bent state before encapsulation. (**d**) Optical image showing the f-BUI encapsulated in silicone. (**e**) Micro-CT 3D rendering image of a rat brain after implantation of f-BUI on day 30.

**Table 1 micromachines-15-00982-t001:** Types, applications, and characteristics of common sensors.

Types of Sensors	Main Applications	Advantages	Disadvantages
piezoelectric [13,14,16]	weapon guidance, industrial inspection, ocean exploration, medical diagnosis	strong self-powering, capability, rapid response speed, flexible structure, excellent transmitting/receiving performance	limited frequency response, weak direct current response, high cost
piezoresistive [18]	pressure sensing, motion monitoring	easy integration, simple structure, low power consumption	weak temperature stability, complex fabrication process, susceptible to external interference
Magnetostrictive [19,20]	liquid level measurement acoustic measurement, structural monitoring	high precision, large measuring range, fast response speed	weak anti-interference capability, short product lifespan, complex signal processing
Capacitive [20]	pressure sensing, industrial inspection, motion monitoring	high resolution, fast response speed, great dynamic response	complex power supply system, high maintenance requirements, high cost
Inductive [20,21]	displacement measurement, humidity measurement, non-destructive testing	high sensitivity, strong durability, great dynamic response	weak anti-interference capability, high-temperature error, large volume

**Table 2 micromachines-15-00982-t002:** Summary of properties, characteristics, and applications of piezoelectric composites prepared by different preparation processes.

Preparation Process Type	Literature Source	Piezoelectric Strain Coefficient	Bending Radius	Electro- Mechanical Coupling Coefficient	Fracture Stress	Fracture Strain	Maximum Output Voltage	Maximum Output Current	Advantages	Disadvantages	Potential Applications
Molding by Hot Pressing	He [101]	~45pC/N	/	~0.12	/	/	/	/	Simple process, suitable for mass production	High thermal stability requirements, weak piezoelectric performance.	Actuators, energy harvesters.
Electro- spinning	Ali [70]	/	/	/	30~40 MPa	~5.0%	4.42 V	/	High precision, can produce flexible materials with high piezoelectric properties	Low yield, high equipment operation requirements.	Wearable devices, implantable devices.
	Su [107]	/	/	/	35 MPa	2.25~4.3%	200 V	0.5 μA			
	Chen [108]	/	/	/	/	/	88 V	5.85 μA			
Electrospray Deposition	Li [110]	560 pm/V	/	/	/	/	30 V	/	Ultra-fast preparation efficiency and precision; excellent piezoelectric film performance	Not suitable for developing large-sized materials, relatively complex process.	Wearable devices, energy harvesters.
Dice-Fill	Hou [85]	/	30 mm	0.74	/	/	/	/	Simple process, suitable for developing large-sized composite materials	Low processing precision, not suitable for developing small-sized materials.	Underwater acoustic transducers, and ultrasonic transducers.
	Hao [86]	464.3 pC/N	150 mm	0.70	/	/	/	/			
Injection- Molding	Zhang [114]	530 pC/N	/	0.65	/	/	/	/	Simple process, developing composites with high aspect ratios or complex structures.	High cost and high mold design requirements.	Ultrasonic transducers, medical diagnostic devices.
Freeze- Casting	Yan [117]	354~434 pC/N	/	/	/	/	30.2 V	13.8 μA	Suitable for developing porous ceramic-based composites.	Long process cycle, high mold requirements.	Energy harvesters, tactile sensors.
	Xie [118]	750 pC/N	8 mm	/	/	/	1.2 V	/			
3D Printing	Zeng [121]	60 pC/N	/	~0.31	/	/	0.18 V	/	Developing composites with complex structures, highly adaptable.	High requirements for materials and equipment.	Micro ultrasonic transducers, bionic sensors.
	Li [123]	153 pC/N	/	/	5.5 MPa	/	6 V	30.18 μA			

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
