# Peer review of "Flexible Electronics: Advancements and Applications of Flexible Piezoelectric Composites in Modern Sensing Technologies"

_micromachines, 2024, doi:10.3390/mi15080982_

Round 1
Reviewer 1 Report
Comments and Suggestions for Authors
In this manuscript, Zhang et al. present a nice review of the recent developments of flexible piezoelectric composite materials and their applications on advanced sensors and electronics. Furthermore, they systematically discussed the composite types and typical manufacturing methods, as well as their applications across various fields such as underwater detection, electronic skin sensing, wearable sensors, and targeted therapy. Finally, they provided the prospects of using piezoelectric composite materials for next-generation electronics. The publication of this work in Micromachines is recommended after addressing the following comments:
1. Why use the term “Flex-Fit Electronics” in the title of this work? This does not seem to be a commonly used word that will prohibit the quick and easy understanding of readers.
2. The abstract needs further improvement. The authors should introduce the concept of the piezoelectric effect and highlight the unique advantages of piezoelectric sensors, such as their self-powered ability. This will help a broader range of readers to follow the article.
3. The introduction should consider the main advantages and limitations of the piezoelectric sensors in comparison with other types (e.g., piezoresistive, capacitive sensors) reported in the literature.
4. The authors are suggested to provide a brief figure to illustrate the sensing mechanism of piezoelectric materials first.
5. This manuscript should incorporate a comparative table on the piezoelectric coefficients, mechanical properties, main advantages, drawbacks, and potential applications of different piezoelectric composite materials fabricated by different techniques. The objective is to compare them or establish a link between their properties and their performance once integrated into devices.
6. Flexible/stretchable piezoelectric 1-3 composite-based ultrasonic transducers for sensing deep tissues have been recently a hot topic in this field. The authors should report some recent literature on this topic.
7. The fabrication techniques introduced in this manuscript is limited, only including hot pressing, electrospinning, dice-fill, and injection molding. Besides these techniques, some other approaches have also been widely used for the manufacturing of piezoelectric composite materials such as electrospray deposition (1), 3D printing (2-5), freeze casting (6,7). The authors are suggested to discuss these other fabrication techniques.
(1) Li, X. et al. Fast and versatile electrostatic disc microprinting for piezoelectric elements. Nat Commun 14, 6488 (2023).
(2) Li J, Yang F, Long Y, et al. Bulk ferroelectric metamaterial with enhanced piezoelectric and biomimetic mechanical properties from additive manufacturing[J]. ACS nano, 2021, 15(9): 14903-14914.
(3) Sebastian T, Bach M, Geiger A, et al. Investigation of electromechanical properties on 3-d printed piezoelectric composite scaffold structures[J]. Materials, 2021, 14(20): 5927.
(4) Zeng Y, Jiang L, Sun Y, et al. 3D-printing piezoelectric composite with honeycomb structure for ultrasonic devices[J]. Micromachines, 2020, 11(8): 713.
(5) Grinberg D, Siddique S, Le M Q, et al. 4D Printing based piezoelectric composite for medical applications[J]. Journal of Polymer Science Part B: Polymer Physics, 2019, 57(2): 109-115.
(6) Xie M, Zhang Y, Kraśny M J, et al. Flexible and active self-powered pressure, shear sensors based on freeze casting ceramic–polymer composites[J]. Energy & environmental science, 2018, 11(10): 2919-2927.
(7) Fukushima M, Fujiwara T, Fey T, et al. One‐or two‐dimensional channel structures and properties of piezoelectric composites via freeze‐casting[J]. Journal of the American Ceramic Society, 2017, 100(12): 5400-5408.
8. One highlight of this manuscript is the discussion of different types of newly developed piezoelectric composite materials, including PZT-PDMS, PZT-PVDF, and PVDF-ZnO, which are mostly non-degradable and even toxic for organisms. However, the discussion of piezoelectric composite biomaterials, which have received significant attention in recent years, is almost absent in this article. We suggest that the authors consider including a section discussing piezoelectric composite biomaterials in the review. The following recent literatures could be considered (1-5).
(1) Yang, F. et al. Wafer-scale heterostructured piezoelectric bio-organic thin films. Science (1979) 373, 337–342 (2021).
(2) Zhang, Z. et al. Active self-assembly of piezoelectric biomolecular films via synergistic nanoconfinement and in-situ poling. Nat Commun 14, 4094 (2023).
(3) Chorsi, Meysam T., et al. "Highly piezoelectric, biodegradable, and flexible amino acid nanofibers for medical applications." Science Advances 9.24 (2023): eadg6075.
(4) Wang, Z. et al. Self-charging and long-term face masks leveraging low-cost, biodegradable and sustainable piezoelectric nanofiber membrane. Nano Materials Science (2024).
(5) Zhang, Z. et al. Van der Waals exfoliation processed biopiezoelectric submucosa ultrathin films. Advanced Materials 34, 2200864 (2022).
(6) Zhang, H.-Y. et al. Biodegradable ferroelectric molecular crystal with large piezoelectric response. Science (1979) 383, 1492–1498 (2024).

Comments on the Quality of English LanguageAuthor Response
Please see the attachment.

Reviewer 2 Report
Comments and Suggestions for Authors
Require Major revision before acceptance. Here are few comments that can help to improve the quality of the manuscript
Comments
1. As the conductive filler has been intensively used to improve the piezoelectricity of the polymer, the authors should discuss the conductive filler in order to enhance the piezoelectricity for the broader readership. The authors are advised to follow a few relevant papers for better insight on this topic. https://doi.org/10.1021/acsami.0c05819, https://doi.org/10.1016/j.compscitech.2020.108600
2. The authors must show a schematic of the working mechanism of the piezoelectric device, along with the output voltage and current characteristic curve.
3. As percolation and piezoelectricity are closely related due to dielectric relaxation and polarization, the authors should discuss this topic in detail as well.
4. The authors mention the incorporation of piezoelectric ceramics to enhance the piezoelectric property of the polymer. However, the authors should also discuss the proportion of the ceramic filler needed to maintain the optimum output power while considering the elasticity and strength of the polymer. In addition, the authors should discuss the effect of agglomeration on the polarization of the piezoelectric materials.
5. It would be beneficial to add some experimental results and photographs of the samples in the "Fabrication Process of FPC Materials" section for better understanding and to enhance readership.
Round 2
Reviewer 2 Report
Comments and Suggestions for Authors
The authors have successfully addressed all of my concerns and revised the manuscript well. Thus, I would like to recommend accepting the manuscript in its current state.